# Thompson Sampling with Information Relaxation Penalties

**Seungki Min**
Columbia Business School

**Costis Maglaras**
Columbia Business School

**Ciamac C. Moallemi**
Columbia Business School

## Abstract

We consider a finite-horizon multi-armed bandit (MAB) problem in a Bayesian setting, for which we propose an *information relaxation sampling* framework. With this framework, we define an intuitive family of control policies that include Thompson sampling (TS) and the Bayesian optimal policy as endpoints. Analogous to TS, which, at each decision epoch pulls an arm that is best with respect to the randomly sampled parameters, our algorithms sample entire future reward realizations and take the corresponding best action. However, this is done in the presence of "penalties" that seek to compensate for the availability of future information.

We develop several novel policies and performance bounds for MAB problems that vary in terms of improving performance and increasing computational complexity between the two endpoints. Our policies can be viewed as natural generalizations of TS that simultaneously incorporate knowledge of the time horizon and explicitly consider the exploration-exploitation trade-off. We prove associated structural results on performance bounds and suboptimality gaps. Numerical experiments suggest that this new class of policies perform well, in particular in settings where the finite time horizon introduces significant exploration-exploitation tension into the problem.

## 1 Introduction

Dating back to the earliest work [2, 10], multi-armed bandit (MAB) problems have been considered within a Bayesian framework, in which the unknown parameters are modeled as random variables drawn from a known prior distribution. In this setting, the problem can be viewed as a Markov decision process (MDP) with a state that is an information state describing the beliefs of unknown parameters that evolve stochastically upon each play of an arm according to Bayes' rule.

Under the objective of expected performance, where the expectation is taken with respect to the prior distribution over unknown parameters, the (Bayesian) optimal policy (OPT) is characterized by Bellman equations immediately following from the MDP formulation. In the discounted infinite-horizon setting, the celebrated Gittins index [10] determines an optimal policy, despite the fact that its computation is still challenging. In the non-discounted finite-horizon setting, which we consider, the problem becomes more difficult [1], and except for some special cases, the Bellman equations are neither analytically nor numerically tractable, due to the curse of dimensionality. In this paper, we focus on the Bayesian setting, and attempt to apply ideas from dynamic programming (DP) to develop tractable policies with good performance.

To this end, we apply the idea of *information relaxation* [4], a technique that provides a systematic way of obtaining the performance bounds on the optimal policy. In multi-period stochastic DP

problems, admissible policies are required to make decisions based only on previously revealed information. The idea of information relaxation is to consider non-anticipativity as a constraint imposed on the policy space that can be relaxed, while simultaneously introducing a penalty for this relaxation into the objective, as in the usual Lagrangian relaxations of convex duality theory. Under such a relaxation, the decision maker (DM) is allowed to access future information and is asked to solve an optimization problem so as to maximize her total reward, in the presence of penalties that punish any violation of the non-anticipativity constraint. When the penalties satisfy a condition (dual feasibility, formally defined in §3), the expected value of the maximal reward adjusted by the penalties provides an upper bound on the expected performance of the (non-anticipating) optimal policy.

The idea of relaxing the non-anticipativity constraint has been studied in different contexts [17, 6, 18, 11], and was later formulated as a formal framework by [4], upon which our methodology is developed. This framework has been applied to a variety of applications including optimal stopping problems [7], linear-quadratic control [12], dynamic portfolio execution [13], and more (see [3]). Typically, the application of this method to a specific class of MDPs requires custom analysis. In particular, it is not always easy to determine penalty functions that (1) yield a relaxation that is tractable to solve, and (2) provide tight upper bounds on the performance of the optimal policy. Moreover, the established information relaxation theory focuses on upper bounds and provides no guidance on the development of tractable policies.

Our contribution is to apply the information relaxation techniques to the finite-horizon stochastic MAB problem, explicitly exploiting the structure of a Bayesian learning process. In particular,

1. we propose a series of information relaxations and penalties of increasing computational complexity;
2. we systematically obtain the upper bounds on the best achievable expected performance that trade off between tightness and computational complexity;
3. and we develop associated (randomized) policies that generalize Thompson sampling (TS) in the finite-horizon setting.

In our framework, which we call *information relaxation sampling*, each of the penalty functions (and information relaxations) determines one policy and one performance bound given a particular problem instance specified by the time horizon and the prior beliefs. As a base case for our algorithms, we have TS [21] and the conventional regret benchmark that has been used for Bayesian regret analysis since [15]. At the other extreme, the optimal policy OPT and its expected performance follow from the "ideal" penalty (which, not surprisingly, is intractable to compute). By picking increasingly strict information penalties, we can improve the policy and the associated bound between the two extremes of TS and OPT.

As an example, one of our algorithms, IRS.FH, provides a very simple modification of TS that naturally incorporates time horizon $T$. Recalling that TS makes a decision based on sampled parameters from the posterior distribution in each epoch, we focus on the fact that knowing the parameters is as informative as having an infinite number of future reward observations in terms of the best arm identification. By contrast, IRS.FH makes a decision based on future Bayesian estimates, updated with only $T - 1$ future reward realizations for each arm, where the rewards are sampled based on the prior belief at the moment. When $T = 1$ (equivalently, at the last decision epoch), such a policy takes a myopically best action based only on the current estimates, which is indeed an optimal decision, whereas TS would still explore unnecessarily. While keeping the recursive structure of the sequential decision-making process of TS, IRS.FH naturally performs less exploration than TS as the remaining time horizon diminishes. This mitigates a common practical criticism of TS: it explores too much.

Beyond this, we propose other algorithms that more explicitly quantify the benefit of exploration and more explicitly trade off between exploration and exploitation, at the cost of additional computational complexity. As we increase the complexity, we achieve policies that improve performance, and separately provide tighter tractable computational upper bounds on the expected performance of any policy for a particular problem instance. By providing natural generalizations of TS, our work provides both a deeper understanding of TS and improved policies that do not require tuning. Since TS has been shown to be asymptotically regret optimal [5], our improvements can at best be (asymptotically) constant factor improvements by that metric. On the other hand, TS is extremely

popular in practice, and we demonstrate in numerical examples that the improvements can be significant and are likely to be of practical interest.

Moreover, we develop upper bounds on performance that are useful in their own right. Suppose that a decision maker faces a particular problem instance and is considering any particular MAB policy (be it one we suggest or otherwise). By simulating the policy, a lower bound on the performance of the optimal policy can be found. We introduce a series of upper bounds that can also be evaluated in any problem instance via simulation. Paired with the lower bound, these provide a computational, simulation-based "confidence interval" that can be helpful to the decision maker. For example, if the upper bound and lower bound are close, the suboptimality gap of the policy under consideration is guaranteed to be small, and it is not worth investing in better policies.

## 2   Notation and Preliminaries

**Problem.** We consider a classical stochastic MAB problem with $K$ *independent arms* and *finite-horizon $T$*. At each decision epoch $t = 1, \dots, T$, the decision maker (DM) pulls an arm $a_t \in \mathcal{A} \triangleq \{1, \dots, K\}$ and earns a *stochastic reward* associated with arm $a_t$. More formally, the reward from the $n^{\text{th}}$ pull of arm $a$ is denoted by $R_{a,n}$ which is independently drawn from unknown distribution $\mathcal{R}_a(\theta_a)$, where $\theta_a \in \Theta_a$ is the *parameter* associated with arm $a$. We also have a prior distribution $\mathcal{P}_a(y_a)$ over unknown parameter $\theta_a$, where $y_a \in \mathcal{Y}_a$, which we call *belief*, is a hyperparameter describing the prior distribution: $\theta_a \sim \mathcal{P}_a(y_a)$ and $R_{a,n}|\theta_a \sim \mathcal{R}_a(\theta_a)$ for all $n \in \mathbb{N}$ and all $a \in \mathcal{A}$.

We define the *outcome* $\omega \triangleq ((\theta_a)_{a \in \mathcal{A}}, (R_{a,n})_{a \in \mathcal{A}, n \in \mathbb{N}})$ that incorporates the all uncertainties that the DM encounters. Given the prior belief vector $\mathbf{y} \triangleq (y_1, \dots, y_K) \in \mathcal{Y}$, we let $\mathcal{I}(\mathbf{y})$ be the prior distribution of outcome $\omega$ that would be described with $\mathcal{P}_a$'s and $\mathcal{R}_a$'s.

We additionally define the true mean reward $\mu_a$ and its Bayesian estimate $\hat{\mu}_{a,n}$ as follows

$$\mu_a(\theta_a) \triangleq \mathbb{E}\left[R_{a,n}|\theta_a\right], \quad \hat{\mu}_{a,n}(\omega; y_a) \triangleq \mathbb{E}\left[\mu_a(\theta_a)|R_{a,1}, \dots, R_{a,n}\right]. \tag{1}$$

Through out the paper, we assume that the rewards are absolutely integrable over the prior distribution: i.e., $\mathbb{E}\left[|R_{a,n}|\right] < \infty$, or more explicitly, $\mathbb{E}_{r \sim \mathcal{R}_a(\mathcal{P}_a(y_a))}\left[|r|\right] < \infty$ where $\mathcal{R}_a(\mathcal{P}_a(y_a))$ denotes the (unconditional) distribution of reward $R_{a,n}$ as a doubly stochastic random variable.

**Policy.** Given an action sequence up to time $t$, $\mathbf{a}_{1:t} \triangleq (a_1, \dots, a_t) \in \mathcal{A}^t$, define the number of pulls $n_t(\mathbf{a}_{1:t}, a) \triangleq \sum_{s=1}^{t} \mathbf{1}\{a_s = a\}$ for each arm $a$, and the corresponding reward realization $r_t(\mathbf{a}_{1:t}, \omega) \triangleq R_{a_t, n_t(\mathbf{a}_{1:t}, a_t)}$. The *natural filtration* $\mathcal{F}_t(\mathbf{a}_{1:t}, \omega; T, \mathbf{y}) \triangleq \sigma\left(T, \mathbf{y}, (a_s, r_s(\mathbf{a}_{1:s}, \omega))_{s \in [t]}\right)$ encodes the observations revealed up to time $t$ (inclusive).

Let $\mathbf{a}_{1:t}^{\pi}$ be the action sequence taken by a policy $\pi$. A policy $\pi$ is called *non-anticipating* if its every action $a_t^{\pi}$ is $\mathcal{F}_{t-1}$-measurable, and we define $\Pi_{\mathbb{F}}$ be a set of all non-anticipating policies, including randomized ones. The (Bayesian) *performance* of a policy $\pi$ is defined as the expected total reward over the randomness associated with the outcome, i.e.,

$$V(\pi, T, \mathbf{y}) \triangleq \mathbb{E}_{\omega \sim \mathcal{I}(\mathbf{y})}\left[\sum_{t=1}^{T} r_t(\mathbf{a}_{1:t}^{\pi}, \omega)\right]. \tag{2}$$

**MDP formulation.** We assume that we are equipped with a *Bayesian update function* $\mathcal{U} : \mathcal{Y} \times \mathcal{A} \times \mathbb{R} \mapsto \mathcal{Y}$ so that after observing $R_{a,1} = r$ from some arm $a$, the belief vector is updated from $\mathbf{y}$ to $\mathcal{U}(\mathbf{y}, a, r)$ according to Bayes' rule, where only the $a^{\text{th}}$ component is updated in this step.

In a Bayesian framework, the MAB problem has a recursive structure. Given a time horizon $T$ and prior belief $\mathbf{y}$, suppose the DM had just earned $r$ by pulling an arm $a$ at time $t = 1$. The remaining problem for the DM is equivalent to a problem with time horizon $T - 1$ and prior belief $\mathcal{U}(\mathbf{y}, a, r)$. Following from this Markovian structure, we obtain the Bellman equations for the MAB problem:

$$Q^*(T, \mathbf{y}, a) \triangleq \mathbb{E}_{r \sim \mathcal{R}_a(\mathcal{P}_a(y_a))}\left[r + V^*(T - 1, \mathcal{U}(\mathbf{y}, a, r))\right], \quad V^*(T, \mathbf{y}) \triangleq \max_{a \in \mathcal{A}} Q^*(T, \mathbf{y}, a), \tag{3}$$

with $V^*(0, \mathbf{y}) \triangleq 0$ for all $\mathbf{y} \in \mathcal{Y}$. While the Bellman equation is intractable to analyze, it offers a characterization of the Bayesian optimal policy (OPT) and the best achievable performance $V^*$: i.e., $V^*(T, \mathbf{y}) = V(\text{OPT}, T, \mathbf{y}) = \sup_{\pi \in \Pi_{\mathbb{F}}} V(\pi, T, \mathbf{y})$.

# 3 Information Relaxation Sampling

We propose a general framework, which we refer to as *information relaxation sampling* (IRS), that takes as an input a "penalty function" $z_t(\cdot)$, and produces as outputs a policy $\pi^z$ and an associated performance bound $W^z$.

**Information relaxation penalties and inner problem.** If we relax the non-anticipativity constraint imposed on policy space $\Pi_{\mathbb{F}}$ (i.e., $a_t^\pi$ is $\mathcal{F}_{t-1}$-measurable), the DM will be allowed to first observe all future outcomes in advance, and then pick an action (i.e., $a_t^\pi$ is $\sigma(\omega)$-measurable). To compensate for this relaxation, we impose a penalty on the DM for violating the nonanticipativity constraint.

We introduce a *penalty function* $z_t(\mathbf{a}_{1:t}, \omega; T, \mathbf{y})$ to denote the penalty that the DM incurs at time $t$, when taking an action sequence $\mathbf{a}_{1:t}$ given a particular instance specified by $\omega$, $T$ and $\mathbf{y}$. The clairvoyant DM can find the best action sequence that is optimal for a particular outcome $\omega$ in the presence of penalties $z_t$, by solving the following (deterministic) optimization problem, referred to as the *inner problem*:

$$\text{maximize}_{\mathbf{a}_{1:T} \in \mathcal{A}^T} \quad \sum_{t=1}^{T} r_t(\mathbf{a}_{1:t}, \omega) - z_t(\mathbf{a}_{1:t}, \omega; T, \mathbf{y}). \tag{$*$}$$

**Definition 1** (Dual feasibility). *A penalty function $z_t$ is dual feasible if it is ex-ante zero-mean, i.e.,*

$$\mathbb{E}\left[z_t(\mathbf{a}_{1:t}, \omega; T, \mathbf{y}) \,|\, \mathcal{F}_{t-1}(\mathbf{a}_{1:t-1}, \omega; T, \mathbf{y})\right] = 0, \quad \forall \mathbf{a}_{1:t} \in \mathcal{A}^t, \quad \forall t \in [T]. \tag{4}$$

To clarify the notion of conditional expectation, we remark that the mapping $\mathbf{a}_{1:t} \mapsto z_t(\mathbf{a}_{1:t}, \omega; T, \mathbf{y})$ is a stochastic function of the action sequence $\mathbf{a}_{1:t}$ since the outcome $\omega$ is random.[1] The dual feasibility condition requires that the DM who makes decisions on the natural filtration will receive zero penalties in expectation.

**IRS performance bound.** Let $W^z(T, \mathbf{y})$ be the expected maximal value of the inner problem $(*)$, when the outcome $\omega$ is randomly drawn from its prior distribution $\mathcal{I}(\mathbf{y})$, i.e., the expected total payoff that a clairvoyant DM can achieve in the presence of penalties:

$$W^z(T, \mathbf{y}) \triangleq \mathbb{E}_{\omega \sim \mathcal{I}(\mathbf{y})}\left[\max_{\mathbf{a}_{1:T} \in \mathcal{A}^T}\left\{\sum_{t=1}^{T} r_t(\mathbf{a}_{1:t}, \omega) - z_t(\mathbf{a}_{1:t}, \omega; T, \mathbf{y})\right\}\right]. \tag{5}$$

We can obtain this value numerically via simulation: draw outcomes $\omega^{(1)}, \omega^{(2)}, \ldots, \omega^{(S)}$ independently from $\mathcal{I}(\mathbf{y})$, solve the inner problem for each outcome separately, and then take the average of the maximal values across these samples. The following theorem shows that $W^z$ is indeed a valid performance bound of the stochastic MAB problem.

**Theorem 1** (Weak duality and strong duality). *If the penalty function $z_t$ is dual feasible, $W^z$ is an upper bound on the optimal value $V^*$: for any $T$ and $\mathbf{y}$,*

$$\text{(Weak duality)} \qquad W^z(T, \mathbf{y}) \geq V^*(T, \mathbf{y}). \tag{6}$$

*There exists a dual feasible penalty function, referred to as the ideal penalty $z_t^{\text{ideal}}$, such that*

$$\text{(Strong duality)} \qquad W^{\text{ideal}}(T, \mathbf{y}) = V^*(T, \mathbf{y}). \tag{7}$$

*The ideal penalty function $z_t^{\text{ideal}}$ has a following functional form:*

$$\begin{aligned}
z_t^{\text{ideal}}(\mathbf{a}_{1:t}, \omega) &\triangleq r_t(\mathbf{a}_{1:t}, \omega) - \mathbb{E}\left[r_t(\mathbf{a}_{1:t}, \omega) \,|\, \mathcal{F}_{t-1}(\mathbf{a}_{1:t-1}, \omega)\right] \\
&\quad + V^*\left(T - t, \mathbf{y}_t(\mathbf{a}_{1:t}, \omega)\right) - \mathbb{E}\left[V^*\left(T - t, \mathbf{y}_t(\mathbf{a}_{1:t}, \omega)\right) \,|\, \mathcal{F}_{t-1}(\mathbf{a}_{1:t-1}, \omega)\right].
\end{aligned} \tag{8}$$

A good penalty function precisely penalizes for the additional profit extracted from using the future information $\omega$. At extreme, the ideal penalty $z_t^{\text{ideal}}$, intractable however, removes any incentive to deviate from OPT and results in the strong duality. In (8), $\mathbf{y}_t(\mathbf{a}_{1:t}, \omega)$ represents the posterior belief that the DM would have at time $t$ after observing the reward realizations associated with $\mathbf{a}_{1:t}$ given $\omega$.

**IRS policy.** Given a penalty function $z_t$, we characterize a randomized and non-anticipating IRS policy $\pi^z \in \Pi_\mathbb{F}$ as follows. The policy $\pi^z$ specifies "which arm to pull when the remaining time is $T$ and current belief is $\mathbf{y}$." Given $T$ and $\mathbf{y}$, it (i) first samples an outcome $\tilde{\omega}$ from $\mathcal{I}(\mathbf{y})$ randomly, (ii) solves the inner problem to find a best action sequence $\tilde{\mathbf{a}}_{1:T}^*$ with respect to $\tilde{\omega}$ in the presence of penalties $z_t$, and (iii) takes the first action $\tilde{a}_1^*$ that the clairvoyant optimal solution $\tilde{\mathbf{a}}_{1:T}^*$ suggests. Analogous to Thompson sampling, **it repeats steps (i)–(iii) at every decision epoch**, while updating the remaining time $T$ and belief $\mathbf{y}$ upon each reward realization.

---

**Algorithm 1:** Information relaxation sampling (IRS) policy

**Function** IRS$(T, \mathbf{y}; z)$

1    Sample $\tilde{\omega} \sim \mathcal{I}(\mathbf{y})$ (equivalently, $\tilde{\theta}_a \sim \mathcal{P}_a(y_a)$ and $\tilde{R}_{a,n} \sim \mathcal{R}_a(\tilde{\theta}_a)$, $\forall a \in \mathcal{A}$, $\forall n \in [T]$)

2    Find the best action sequence with respect to $\tilde{\omega}$ under penalties $z_t$:

$$\tilde{\mathbf{a}}_{1:T}^* \leftarrow \operatorname{argmax}_{\mathbf{a}_{1:T} \in \mathcal{A}^T} \left\{ \sum_{t=1}^T r_t(\mathbf{a}_{1:t}, \tilde{\omega}) - z_t(\mathbf{a}_{1:t}, \tilde{\omega}; T, \mathbf{y}) \right\}$$

3    **return** $\tilde{a}_1^*$

**Procedure** IRS-Outer$(T, \mathbf{y}; z)$

1    $\mathbf{y}_0 \leftarrow \mathbf{y}$

2    **for** $t = 1, 2, \ldots, T$ **do**

3      Play $a_t \leftarrow$ IRS$(T - t + 1, \mathbf{y}_{t-1}; z)$

4      Earn and observe a reward $r_t$ and update belief $\mathbf{y}_t \leftarrow \mathcal{U}(\mathbf{y}_{t-1}, a_t, r_t)$

   **end**

---

**Remark 1.** *The ideal penalty yields the Bayesian optimal policy: i.e., $V(\pi^{\text{ideal}}, T, \mathbf{y}) = V^*(T, \mathbf{y})$.*

**Choice of penalty functions.** IRS policies include Thompson sampling and the Bayesian optimal policy as two extremal cases. We propose a set of penalty functions spanning these two. While deferring the detailed explanations in §3.1 – §3.4, we briefly list the penalty functions:

$$z_t^{\text{TS}}(\mathbf{a}_{1:t}, \omega) \triangleq r_t(\mathbf{a}_{1:t}, \omega) - \mathbb{E}\left[ r_t(\mathbf{a}_{1:t}, \omega) \,|\, \theta_1, \ldots, \theta_K \right] \tag{9}$$

$$z_t^{\text{IRS.FH}}(\mathbf{a}_{1:t}, \omega) \triangleq r_t(\mathbf{a}_{1:t}, \omega) - \mathbb{E}\left[ r_t(\mathbf{a}_{1:t}, \omega) \,|\, \hat{\mu}_{1,T-1}(\omega), \ldots, \hat{\mu}_{K,T-1}(\omega) \right] \tag{10}$$

$$z_t^{\text{IRS.V-ZERO}}(\mathbf{a}_{1:t}, \omega) \triangleq r_t(\mathbf{a}_{1:t}, \omega) - \mathbb{E}\left[ r_t(\mathbf{a}_{1:t}, \omega) \,|\, \mathcal{F}_{t-1}(\mathbf{a}_{1:t-1}, \omega) \right] \tag{11}$$

$$z_t^{\text{IRS.V-EMAX}}(\mathbf{a}_{1:t}, \omega) \triangleq r_t(\mathbf{a}_{1:t}, \omega) - \mathbb{E}\left[ r_t(\mathbf{a}_{1:t}, \omega) \,|\, \mathcal{F}_{t-1}(\mathbf{a}_{1:t-1}, \omega) \right] \tag{12}$$
$$+ W^{\text{TS}}\left( T - t, \mathbf{y}_t(\mathbf{a}_{1:t}, \omega) \right) - \mathbb{E}\left[ W^{\text{TS}}\left( T - t, \mathbf{y}_t(\mathbf{a}_{1:t}, \omega) \right) \,\middle|\, \mathcal{F}_{t-1}(\mathbf{a}_{1:t-1}, \omega) \right]$$

To help understanding, we provide an identity as an example: $\mathbb{E}\left[ r_t(\mathbf{a}_{1:t}, \omega) \,|\, \mathcal{F}_{t-1}(\mathbf{a}_{1:t-1}, \omega) \right] = \mathbb{E}\left[ \mu_{a_t}(\theta_{a_t}) \,\middle|\, R_{a_t,1}, \ldots, R_{a_t, n_{t-1}(\mathbf{a}_{1:t-1}, a_t)} \right] = \hat{\mu}_{a_t, n_{t-1}(\mathbf{a}_{1:t-1}, a_t)}(\omega)$ – they all represent the mean reward that the DM expects to get from arm $a_t$ right before making a decision at time $t$.

**Remark 2.** *All penalty functions (8)–(12) are dual feasible.*

As we sequentially increase its complexity, from $z^{\text{TS}}$ to $z^{\text{ideal}}$, the penalty function more accurately penalizes the benefit of knowing the future outcomes, more explicitly preventing the DM from exploiting the future information. As summarized in Table 1, it makes the inner problem closer to the original stochastic optimization problem that results in a better performing policy and a tighter performance bound. As a result, we achieve a family of algorithms that are intuitive and tractable, exhibiting a trade-off between quality and computational efficiency.

### 3.1 Thompson Sampling

With the penalty function $z_t^{\text{TS}}(\mathbf{a}_{1:t}, \omega) = r_t(\mathbf{a}_{1:t}, \omega) - \mu_{a_t}(\theta_{a_t})$, the inner problem $(*)$ reduces to

$$\max_{\mathbf{a}_{1:T} \in \mathcal{A}^T} \left\{ \sum_{t=1}^T r_t(\mathbf{a}_{1:t}, \omega) - z_t^{\text{TS}}(\mathbf{a}_{1:t}, \omega) \right\} = \max_{\mathbf{a}_{1:T} \in \mathcal{A}^T} \left\{ \sum_{t=1}^T \mu_{a_t}(\theta_{a_t}) \right\} = T \times \max_{a \in \mathcal{A}} \mu_a(\theta_a). \tag{13}$$

The resulting performance bound $W^{\text{TS}}(T, \mathbf{y})$ is $\mathbb{E}\left[ T \times \max_{a \in \mathcal{A}} \mu_a(\theta_a) \right]$ that is the conventional benchmark in a Bayesian setting [15, 19]. The corresponding IRS policy $\pi^{\text{TS}}$ restores Thompson sampling: when the sampled outcome $\tilde{\omega}$ is used instead, it plays the arm $\tilde{a}_1^* = \operatorname{argmax}_a \mu_a(\tilde{\theta}_a)$ where each $\tilde{\theta}_a$ is sampled from $\mathcal{P}_a(y_a)$. Recall that this sampling-based decision making is repeated in each epoch, while updating the belief sequentially, as described in IRS-OUTER in Algorithm 1.

| Penalty function | Policy | Performance bound | Inner problem | Run time |
|---|---|---|---|---|
| $z_t^{\text{TS}}$ | TS | $W^{\text{TS}}$ | Find a best arm given parameters. | $O(K)$ |
| $z_t^{\text{IRS.FH}}$ | $\pi^{\text{IRS.FH}}$ | $W^{\text{IRS.FH}}$ | Find a best arm given finite observations. | $O(K)^\dagger$ or $O(KT)$ |
| $z_t^{\text{IRS.V-ZERO}}$ | $\pi^{\text{IRS.V-ZERO}}$ | $W^{\text{IRS.V-ZERO}}$ | Find an optimal allocation of $T$ pulls. | $O(KT^2)$ |
| $z_t^{\text{IRS.V-EMAX}}$ | $\pi^{\text{IRS.V-EMAX}}$ | $W^{\text{IRS.V-EMAX}}$ | Find an optimal action sequence. | $O(KT^K)$ |
| $z_t^{\text{ideal}}$ | OPT | $V^*$ | Solve Bellman equations. | – |

Table 1: List of algorithms associated with the penalty functions (8)–(12). Run time represents the time complexity of solving one instance of inner problem, that is, the time required to obtain one sample of performance bound $W^z$ or to make a single decision in policy $\pi^z$. $^\dagger$In IRS.FH, $O(K)$ is achievable when the prior distribution $\mathcal{P}_a$ is a conjugate prior of the reward distribution $\mathcal{R}_a$.

## 3.2   IRS.FH

Recall that $\hat{\mu}_{a,T-1}(\omega)$ is the Bayesian estimate on the mean reward of an arm $a$ inferred from the first $T-1$ reward realizations $R_{a,1}, \ldots, R_{a,T-1}$. Given (10), the optimal solution to the inner problem $(*)$ is to pull an arm with the highest $\hat{\mu}_{a,T-1}(\omega)$ from beginning to the end:

$$\max_{\mathbf{a}_{1:T} \in \mathcal{A}^T} \left\{ \sum_{t=1}^{T} r_t(\mathbf{a}_{1:t}, \omega) - z_t^{\text{IRS.FH}}(\mathbf{a}_{1:t}, \omega) \right\} = \max_{\mathbf{a}_{1:T} \in \mathcal{A}^T} \left\{ \sum_{t=1}^{T} \hat{\mu}_{a_t,T-1}(\omega) \right\} = T \times \max_{a \in \mathcal{A}} \hat{\mu}_{a,T-1}(\omega).$$
(14)

IRS.FH is almost identical to TS except that $\mu_a(\theta_a)$ is replaced with $\hat{\mu}_{a,T-1}(\omega)$. Note that $\hat{\mu}_{a,T-1}(\omega)$ is less informative than $\mu_a(\theta_a)$ for the DM, since she will never be able to learn $\mu_a(\theta_a)$ perfectly within a finite horizon. In terms of estimation, knowing the parameters is equivalent to having the infinite number of observations. The inner problem of TS asks the DM to "identify the best arm based on the infinite number of samples" whereas that of IRS.FH asks her to "identify the best arm based on the finite number of samples," which takes into account the length of time horizon explicitly.

Focusing on the policies $\pi^{\text{IRS.FH}}$ and $\pi^{\text{TS}}$ (where the randomly generated $\mu_a(\tilde{\theta}_a)$ and $\hat{\mu}_{a,T-1}(\tilde{\omega})$ are used), we observe that the distribution of $\hat{\mu}_{a,T-1}(\tilde{\omega})$ will be more concentrated while both have the same mean $\bar{\mu}_a \triangleq \mathbb{E}[\mu_a(\tilde{\theta}_a)] = \hat{\mu}_{a,0}$. Since the variance of $\hat{\mu}_{a,T-1}(\tilde{\omega})$ and $\mu_a(\tilde{\theta}_a)$ govern the degree of random exploration (deviating from the myopic decision of pulling an arm with the largest $\bar{\mu}_a$), $\pi^{\text{IRS.FH}}$ naturally explores less than TS, in particular when it approaches the end of the horizon ($T \searrow 1$). For the performance bounds, by the same reason, we have $W^{\text{IRS.FH}} = \mathbb{E}[T \times \max_a \hat{\mu}_{a,T-1}(\omega)] \leq W^{\text{TS}} = \mathbb{E}[T \times \max_a \mu_a(\theta_a)]$, meaning that IRS.FH yields a performance bound that is tighter than the conventional regret benchmark.

**Sampling $\hat{\mu}_{a,T-1}(\tilde{\omega})$ at once.** In order to obtain $\hat{\mu}_{a,T-1}(\tilde{\omega})$ for a synthesized outcome $\tilde{\omega}$, one may apply Bayes' rule sequentially for each reward realization, which will take $O(KT)$ computations in total. It can be done in $O(K)$ when the belief can be updated in a batch by the use of sufficient statistics. In Beta-Bernoulli and Gaussian MABs, for example, $\hat{\mu}_{a,T-1}(\tilde{\omega})$ can be represented as a convex combination of the current estimate $\bar{\mu}_a$ and the sample mean $\frac{1}{T-1} \sum_{n=1}^{T-1} \tilde{R}_{a,n}$ where $\sum_{n=1}^{T-1} \tilde{R}_{a,n}$ is distributed with Binomial$(T-1, \tilde{\theta}_a)$ for Beta-Bernoulli case and $\mathcal{N}((T-1) \cdot \tilde{\theta}_a, (T-1) \cdot \sigma_a^2)$ for Gaussian case ($\sigma_a^2$ represents the noise variance). After sampling the parameter $\tilde{\theta}_a$, we can sample $\sum_{n=1}^{T-1} \tilde{R}_{a,n}$ directly from the known distribution, then use it to compute $\hat{\mu}_{a,T-1}(\tilde{\omega})$ without sequentially updating the belief. In such cases, a single decision of $\pi^{\text{IRS.FH}}$ can be made within $O(K)$ operations, similar in complexity to TS.

## 3.3   IRS.V-ZERO

Under the penalty $z_t^{\text{IRS.V-ZERO}}$, the DM at time $t$ earns $\mathbb{E}\left[r_t(\mathbf{a}_{1:t}, \omega) \,|\, \mathcal{F}_{t-1}(\mathbf{a}_{1:t-1}, \omega)\right]$, the expected mean reward that she can infer from the observations prior to time $t$. As we defined $R_{a,n}$ to be a reward from the $n^{\text{th}}$ pull on arm $a$ (not the pull at time $n$), the posterior belief associated with each arm is determined only by the number of past pulls on that arm – from the $n^{\text{th}}$ pull on arm $a$, the DM earns $\hat{\mu}_{a,n-1}(\omega)$, irrespective of the detailed sequence of past actions.

Following this observation, solving the inner problem $(*)$ is equivalent to "finding the optimal allocation $(n_1^*, n_2^*, \ldots, n_K^*)$ among $T$ remaining opportunities": omitting $\omega$ for brevity, it reduces to

$$\max_{\mathbf{a}_{1:T} \in \mathcal{A}^T} \left\{ \sum_{t=1}^{T} \hat{\mu}_{a_t, n_{t-1}(\mathbf{a}_{1:t-1}, a_t)} \right\} = \max_{\mathbf{a}_{1:T} \in \mathcal{A}^T} \left\{ \sum_{a=1}^{K} \sum_{n=1}^{n_T(\mathbf{a}_{1:T}, a)} \hat{\mu}_{a, n-1} \right\} = \max_{\mathbf{n}_{1:K} \in N_T} \left\{ \sum_{a=1}^{K} S_{a, n_a} \right\} \tag{15}$$

where $S_{a,n} \triangleq \sum_{m=1}^{n} \hat{\mu}_{a,m-1}$ is the cumulative payoff from the first $n$ pulls of an arm $a$, and $N_T \triangleq \{(n_1, \ldots, n_K) \in \mathbb{Z}_+^K : \sum_{a=1}^{K} n_a = T\}$ is the set of all feasible allocations. Once the $S_{a,n}$'s are computed, this inner problem can be solved within $O(KT^2)$ operations by sequentially applying sup convolution $K$ times. The detailed implementation is provided in Appendix §B.1.

Given an optimal allocation $\tilde{\mathbf{n}}^*$, the policy $\pi^{\text{IRS.V-ZERO}}$ needs to select which arm to pull next. In principle, any arm $a$ that was included in the solution of the inner problem, $\tilde{n}_a^* > 0$, would be fine, but we suggest a selection rule in which the arm that needs most pulls is chosen, i.e., $\operatorname{argmax}_a \tilde{n}_a^*$.

### 3.4 IRS.V-EMAX

Under perfect information relaxation, the DM perfectly knows not only (i) what she will earn at future times but also (ii) how her belief will evolve as a result of her action sequence. The previous algorithms focus on the former component by making the DM to adjust the future rewards by conditioning (e.g., $\mathbb{E}[r_t(a_t)|\boldsymbol{\theta}]$, $\mathbb{E}[r_t(a_t)|\hat{\boldsymbol{\mu}}_{1:K,T-1}]$ and $\mathbb{E}[r_t(a_t)|\mathcal{F}_{t-1}]$). IRS.V-EMAX also focuses on the second component as well by charging an additional cost for using the information on her future belief transitions.

Specifically, the penalty function $z_t^{\text{IRS.V-EMAX}}$ is obtained from $z_t^{\text{ideal}}$ in (8) by replacing $V^*(T, \mathbf{y})$ with $W^{\text{TS}}(T, \mathbf{y})$, which is a tractable alternative. The use of $W^{\text{TS}}(T, \mathbf{y})$ leads to a simple expression for its conditional expectation: since $\boldsymbol{\theta}|\mathcal{F}_{t-1}$ is distributed with $\mathcal{P}(\mathbf{y}_{t-1})$, we have

$$\mathbb{E}\left[W^{\text{TS}}(T-t, \mathbf{y}_t) \big| \mathcal{F}_{t-1}\right] = (T-t) \times \mathbb{E}\left[\max_a \mu_a(\theta_a) \big| \mathcal{F}_{t-1}\right] \tag{16}$$

$$= (T-t) \times \mathbb{E}_{\boldsymbol{\theta} \sim \mathcal{P}(\mathbf{y}_{t-1})}\left[\max_a \mu_a(\theta_a)\right] = W^{\text{TS}}(T-t, \mathbf{y}_{t-1}). \tag{17}$$

We further observe that, given $\omega$, the future belief $\mathbf{y}_t(\mathbf{a}_{1:t}, \omega)$ depends only on how many times each arm has been pulled, irrespective of the sequence of the pulls, and hence, the number of possible future beliefs is $O(T^K)$, not $O(K^T)$. Given the above observations, we can solve the inner problem within $O(KT^K)$ computations by dynamic programming (i.e., by finding a best action at each future belief while iterating over the beliefs in an appropriate order). See §B.2 for details.

## 4 Analysis

**Remark 3** (Single period optimality). *When $T = 1$, all $\pi^{\text{IRS.FH}}$, $\pi^{\text{IRS.V-ZERO}}$, and $\pi^{\text{IRS.V-EMAX}}$ take the optimal action that is pulling the myopically best arm $a^* = \operatorname{argmax}_a \mathbb{E}[\mu_a(\theta_a)]$.*

**Proposition 1** (Asymptotic behavior). *Assume that $\mu_i(\theta_i) \neq \mu_j(\theta_j)$ almost surely for any two distinct arms $i \neq j$. As $T \nearrow \infty$, the distribution of IRS.FH's and IRS.V-ZERO's action[2] converge to that of Thompson sampling: for all $a \in \mathcal{A}$,*

$$\lim_{T \to \infty} \mathbb{P}\left[\text{IRS.FH}(T, \mathbf{y}) = a\right] = \lim_{T \to \infty} \mathbb{P}\left[\text{IRS.V-ZERO}(T, \mathbf{y}) = a\right] = \mathbb{P}\left[\text{TS}(\mathbf{y}) = a\right]. \tag{18}$$

$\text{TS}(\mathbf{y})$, $\text{IRS.FH}(T, \mathbf{y})$ and $\text{IRS.V-ZERO}(T, \mathbf{y})$ denote the action taken by policies $\pi^{\text{TS}}$, $\pi^{\text{IRS.FH}}$ and $\pi^{\text{IRS.V-ZERO}}$, repsectively, when the remaining time is $T$ and the prior belief is $\mathbf{y}$. These are random variables, since each of these policies uses a randomly sampled outcome $\tilde{\omega}$ on its own.

Remark 3 and Proposition 1 state that IRS.FH and IRS.V-ZERO behave like TS during the initial decision epochs, gradually shift toward the myopic scheme and end up with optimal decision; in contrast, TS will continue to explore throughout. The transition from exploration to exploitation under these IRS policies occurs smoothly, without relying on an auxiliary control parameter. While maintaining their recursive structure, IRS policies take into account the horizon $T$, and naturally balance exploitation and exploration.

**Theorem 2** (Monotonicity in performance bounds). IRS.FH *and* IRS.V-ZERO *monotonically improve the performance bound: for any $T$ and* **y**,

$$W^{\text{TS}}(T, \mathbf{y}) \geq W^{\text{IRS.FH}}(T, \mathbf{y}) \geq W^{\text{IRS.V-ZERO}}(T, \mathbf{y}). \qquad (19)$$

*Note that $W^{\text{TS}}(T, \mathbf{y}) = \mathbb{E}_{\boldsymbol{\theta} \sim \mathcal{P}(\mathbf{y})}\left[T \times \max_a \mu_a(\theta_a)\right]$ is the conventional benchmark.*

In addition, we have $W^{\text{IRS.V-EMAX}} \geq W^{\text{ideal}}$ since $W^{\text{ideal}}$ is the lowest attainable upper bound (Theorem 1). Empirically, we also observe $W^{\text{IRS.V-ZERO}} \geq W^{\text{IRS.V-EMAX}}$.

**Theorem 3** (Suboptimality gap). *In the Beta-Bernoulli MAB, for any $T$ and* **y**,

$$W^{\text{TS}}(T, \mathbf{y}) - V(\pi^{\text{TS}}, T, \mathbf{y}) \leq 3K + 2\sqrt{\log T} \times 2\sqrt{KT}, \qquad (20)$$

$$W^{\text{IRS.FH}}(T, \mathbf{y}) - V(\pi^{\text{IRS.FH}}, T, \mathbf{y}) \leq 3K + 2\sqrt{\log T} \times \left(2\sqrt{KT} - \frac{1}{3}\sqrt{T/K}\right), \quad (21)$$

$$W^{\text{IRS.V-ZERO}}(T, \mathbf{y}) - V(\pi^{\text{IRS.V-ZERO}}, T, \mathbf{y}) \leq 2K + \sqrt{\log T} \times \left(2\sqrt{KT} - \frac{1}{3}\sqrt{T/K}\right). \quad (22)$$

We do not have a theoretical guarantee for monotonicity in the actual performance $V(\pi^z, T, \mathbf{y})$ among IRS policies. Instead, Theorem 3 indirectly shows the improvements in suboptmality gap, $W^z(T, \mathbf{y}) - V(\pi^z, T, \mathbf{y})$: although all the bounds have the same asymptotic order of $O(\sqrt{KT \log T})$, the IRS policies improve the leading coefficient or the additional term.

Theorem 2 and 3 highlight that a better choice of penalty function $z_t$ leads to a tighter performance bound $W^z$ and a better performing policy $\pi^z$. Recall that the penalties are designed to penalize the gain of having additional future information. While all IRS algorithms are basically optimistic in a sense that the DM makes a decision believing that the informed outcome ($\omega$ or $\tilde{\omega}$) will be realized, a better penalty function prevents the DM from picking up an action that is overly optimized to a particular future realization.

## 5 Numerical Experiment

We visualize the effectiveness of IRS policies and performance bounds in case of Gaussian MAB with five arms ($K = 5$) with different noise variances. More specifically, each arm $a \in \mathcal{A}$ has the unknown mean reward $\theta_a \sim \mathcal{N}(0, 1^2)$ and yields the stochastic rewards $R_{a,n} \sim \mathcal{N}(\theta_a, \sigma_a^2)$ where $\sigma_1 = 0.1$, $\sigma_2 = 0.4$, $\sigma_3 = 1.0$, $\sigma_4 = 4.0$ and $\sigma_5 = 10.0$. Our experiment includes the state-of-the-art algorithms that are particularly suitable in a Bayesian framework: Bayesian upper confidence bound [14] (BAYES-UCB, with a quantile of $1 - \frac{1}{t}$), information directed sampling [20] (IDS), and optimistic Gittins index [9] (OGI, one-step look ahead approximation with a discount factor $\gamma_t = 1 - \frac{1}{t}$). In the simulation, we randomly generate a set of outcomes $\omega^{(1)}, \ldots, \omega^{(S)}$ and measure the performance of each policy $\pi$, $V(\pi, T, \mathbf{y})$, and the performance bounds, $W^z(T, \mathbf{y})$, via sample average approximation across these sampled outcomes ($S = 20,000$).

Figure 1 plots the regret of policies (solid lines, $W^{\text{TS}}(T, \mathbf{y}) - V(\pi, T, \mathbf{y})$) and the regret bounds (dashed lines, $W^{\text{TS}}(T, \mathbf{y}) - W^z(T, \mathbf{y})$) that are measured at the different values of $T = 5, 10, \ldots, 500$. Our regret measure $W^{\text{TS}} - V(\pi) = \mathbb{E}\left[\sum_{t=1}^{T} \max_a \mu_a(\theta_a) - \mu_{a_t^\pi}(\theta_{a_t^\pi})\right]$ is equivalent to the conventional Bayesian regret [19], and the measure $W^{\text{TS}} - W^z$ provides the lower bound on the achievable regret since $W^{\text{TS}} - V(\pi) \geq W^{\text{TS}} - W^z$ for any policy $\pi \in \Pi_{\mathbb{F}}$ due to the weak duality. Despite the fact that we cannot compute the Bayesian optimal policy directly, we can infer where its regret curve is located in the shaded region of the plot.

Note that lower regret curves are better, and higher bound curves are better. As we incorporate more complicated IRS algorithm from TS to IRS.V-ZERO, we observe a clear improvement in both performances and bounds, as predicted in Theorem 2 and 3. In this particular example, it is crucial to incorporate how much we can learn about each of the arms during the remaining time periods, which heavily depends on the noise level $\sigma_a$ and time horizon $T$. Accordingly, IRS policies outperform to the others, since ours more explicitly incorporate the exploitation-exploration trade-off.

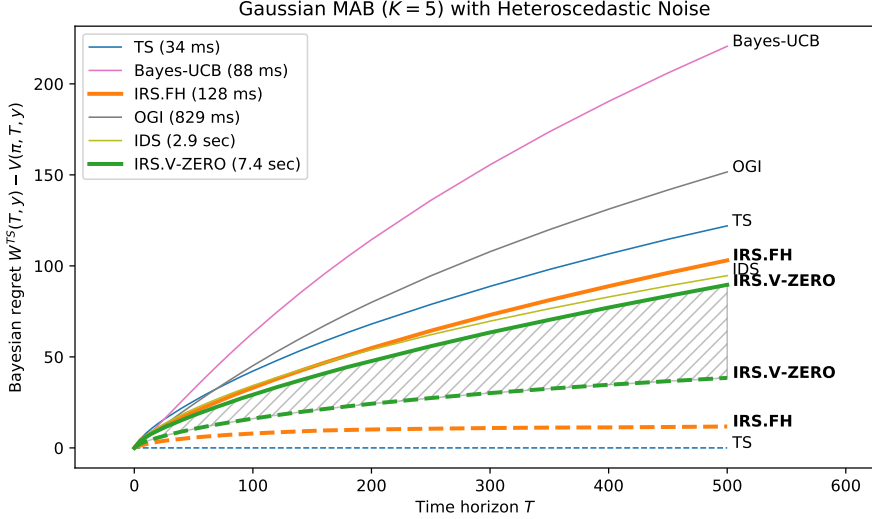

Figure 1: Regret plot for Gaussian MAB with different noise variances. The solid lines represent the (Bayesian) regret of policies, $W^{\text{TS}}(T, \mathbf{y}) - V(\pi, T, \mathbf{y})$, and the dashed lines represent the regret bounds that IRS algorithms produce, $W^{\text{TS}}(T, \mathbf{y}) - W^z(T, \mathbf{y})$. The lowest achievable regret ($W^{\text{TS}}(T, \mathbf{y}) - V^*(T, \mathbf{y})$) should be within the shaded area. The times in the legend represent the average length of time required to simulate each policy for a single problem instance with $T = 500$.

## 6    Discussion

We have developed a unified framework providing a principled method of improving TS that does not require any tuning or additional parameters. Despite the fact that this paper focuses on a finite-horizon MAB with independent arms, the general idea of information relaxation sampling is not restricted to this setting: we briefly illustrate how to extend the framework to a broader class of problems.

**MAB with unknown time horizon.** The framework (penalties, policies, and upper bounds) can naturally incorporate the unknown $T$ within the *Bayesian setting*: i.e., the horizon $T$ is also a random variable whose prior distribution is known. As a simple case, if $T$ is independent of the DM's actions, we can reformulate the objective function of the inner problem as $\sum_{t=1}^{\infty} \gamma_t (r_t(\mathbf{a}_{1:t}, \omega) - z_t(\mathbf{a}_{1:t}, \omega))$ where the discount factor $\gamma_t \triangleq \mathbb{P}[T \geq t]$ is the survival probability, and $r_t(\cdot)$ and $z_t(\cdot)$ are the reward and penalty terms used in the paper. Alternatively, we can treat the random variable $T$ like the random reward realizations – sample $T$ from its prior distribution while a penalty function (additionally) penalizes for the gain from knowing $T$ (one can imagine that the outcome $\omega$ now includes the realization of $T$). Structural results such as weak duality and strong duality will continue to hold.

**MAB in more complicated settings.** Consider the following examples: (i) A finite-horizon MAB with correlated arms (e.g., $R_{a,n} \sim \mathcal{N}(\mathbf{x}_a^\top \boldsymbol{\theta}, \sigma_a^2)$ where $\boldsymbol{\theta} \in \mathbb{R}^d$ is shared across the arms, and $\mathbf{x}_a \in \mathbb{R}^d$ is an arm's feature vector): IRS.V-ZERO can be immediately implemented by adopting the DP algorithm discussed in §B.2. (ii) MAB with the delayed reward realization: IRS.FH can be immediately implemented by simulating the DM's learning process in the presence of delay. (iii) MAB with a budget constraint (in which each arm consumes a certain amount of budget and the DM wants to maximize the total reward within a limited budget. See [8]): all IRS algorithms can be implemented by solving a budget-constrained optimization problem instead of a horizon-constrained optimization problem.

In these extensions, we can obtain not only the online decision making policies but also their performance bounds as in this paper. Generally speaking, our framework provides a systemic way of improving TS by taking into account the exploitation-exploration trade-off more carefully, particularly in the presence of some constraint that incurs incomplete learning. The main challenge would be to design a suitable penalty function that is tractable yet captures the problem-specific exploration-exploitation trade-off precisely.

## Footnotes

[1] As in usual probability theory, $Z(\omega) \triangleq \mathbb{E}[X(\omega)|Y(\omega)]$ represents the expected value of a random variable $X(\omega)$ given the information $Y(\omega)$, and $Z(\omega)$ is itself a random variable that has a dependency on $\omega$.

[2] For IRS.V-ZERO, we assume a particular selection rule such that $\tilde{a}^* = \operatorname{argmax}_a \tilde{n}_a^*$.

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
