[Supplementary Material]

# A  An Illustrating Example

Let us consider a Beta-Bernoulli MAB with $T = 8$ and three arms $(K = 3)$ with the following priors:

$$\theta_1 \sim \text{Beta}(3, 1), \quad \theta_2 \sim \text{Beta}(1, 1), \quad \theta_3 \sim \text{Beta}(1, 3), \tag{23}$$

where $R_{a,n} \sim \text{Bernoulli}(\theta_a)$ for each $a \in \{1, 2, 3\}$ and $n \in \{1, 2, \ldots, 8\}$. Given this prior belief, the expected mean reward of each arm is $\bar{\mu}_1 = \mathbb{E}_{\theta_1 \sim \text{Beta}(3,1)}[\theta_1] = \frac{3}{4}$, $\bar{\mu}_2 = \frac{1}{2}$, and $\bar{\mu}_3 = \frac{1}{4}$, respectively. As an illustrative example, we examine a particular instance where the true outcome $\omega$ is given as follows:

| | **Params** $\theta_a$ | **Rewards** $R_{a,n}$ | | | | | | | |
|---|---|---|---|---|---|---|---|---|---|
| | | $n = 1$ | 2 | 3 | 4 | 5 | 6 | 7 | 8 |
| Arm 1 ($a = 1$) | 0.235 | 0 | 1 | 1 | 1 | 0 | 0 | 0 | 0 |
| Arm 2 ($a = 2$) | 0.443 | 1 | 0 | 0 | 1 | 1 | 1 | 1 | 0 |
| Arm 3 ($a = 3$) | 0.787 | 1 | 1 | 1 | 1 | 0 | 0 | 1 | 1 |

Table 2: An example of outcome in a Beta-Bernoulli MAB with $K = 3$ and $T = 8$.

If we consider only the priors, arm 1 is best since $\bar{\mu}_1$ is largest among $(\bar{\mu}_1, \bar{\mu}_2, \bar{\mu}_3)$. If, however, we have full information about the parameter values, arm 3 is best since $\theta_3$ is largest among $(\theta_1, \theta_2, \theta_3)$.

## A.1  Inner Problems Induced by Different Penalty Functions

**No penalty.** To clarify the role of penalties, we first consider the case of zero penalty ($z_t \equiv 0$), which was not discussed in §3. With zero penalty, the DM at any time earns the current realized reward without adjustment. The clairvoyant DM, who is informed of the outcome $\omega$, can find the best action sequence for this particular outcome $\omega$. Recall that $R_{a,n}$ is defined to be the reward from the $n^{\text{th}}$ pull of arm $a$, not the reward from arm $a$ at time $n$, and so the DM is not allowed to skip any of the reward realizations and the total reward does not depend on the order of pulls. As depicted in the table below, the optimal solution is to pull arm 1 four times, arm 2 once, and arm 3 three times, which yields a total reward of 7.

| | **Payoffs under zero penalty** | | | | | | | | **Maximal payoff** |
|---|---|---|---|---|---|---|---|---|---|
| | $n = 1$ | 2 | 3 | 4 | 5 | 6 | 7 | 8 | |
| Arm 1 | 0 | 1 | 1 | 1 | 0 | 0 | 0 | 0 | |
| Arm 2 | 1 | 0 | 0 | 1 | 1 | 1 | 1 | 0 | 7 |
| Arm 3 | 1 | 1 | 1 | 1 | 0 | 0 | 1 | 1 | |

**TS penalty.** Next, let us examine the penalty $z_t^{\text{TS}}(\mathbf{a}_{1:t}, \omega) \triangleq r_t(\mathbf{a}_{1:t}, \omega) - \mu_{a_t}(\theta_{a_t})$ under which the DM earns $\theta_a$ whenever playing an arm $a$. The hindsight optimal action sequence is to pull arm 3 (the arm with the largest mean reward $\theta_a$) eight times in a row and the DM can earn a total reward of $T \times \theta_3 = 6.296$ at most.

| | **Payoffs under** $z_t^{\text{TS}}$ | | | | | | | | **Maximal payoff** |
|---|---|---|---|---|---|---|---|---|---|
| | $n = 1$ | 2 | 3 | 4 | 5 | 6 | 7 | 8 | |
| Arm 1 | .235 | .235 | .235 | .235 | .235 | .235 | .235 | .235 | |
| Arm 2 | .443 | .443 | .443 | .443 | .443 | .443 | .443 | .443 | 6.296 |
| Arm 3 | .787 | .787 | .787 | .787 | .787 | .787 | .787 | .787 | |

**IRS.FH penalty.** When the penalties are given by $z_t^{\text{IRS.FH}}(\mathbf{a}_{1:t}, \omega) \triangleq r_t(\mathbf{a}_{1:t}, \omega) - \hat{\mu}_{a_t, T-1}(\omega)$, the DM earns $\hat{\mu}_{a, T-1}(\omega)$ whenever playing an arm $a$. Recall that $\hat{\mu}_{a, T-1}(\omega)$ is the Bayesian estimate on mean reward of arm $a$ after observing reward realizations $R_{a,1}, \ldots, R_{a,T-1}$. In this particular example, we have $(\hat{\mu}_{1,T-1}, \hat{\mu}_{2,T-1}, \hat{\mu}_{3,T-1}) = \left(\frac{6}{11}, \frac{6}{9}, \frac{6}{11}\right)$ and the maximal payoff is $T \times \hat{\mu}_{2,T-1} = 5.333$, which can be obtained by playing arm 2 throughout the entire time horizon.

| | Payoffs under $z_t^{\text{IRS.FH}}$ | | | | | | | | Maximal payoff |
|---|---|---|---|---|---|---|---|---|---|
| | $n = 1$ | 2 | 3 | 4 | 5 | 6 | 7 | 8 | |
| Arm 1 | 6/11 | 6/11 | 6/11 | 6/11 | 6/11 | 6/11 | 6/11 | 6/11 | |
| Arm 2 | 6/9 | 6/9 | 6/9 | 6/9 | 6/9 | 6/9 | 6/9 | 6/9 | 5.333 |
| Arm 3 | 6/11 | 6/11 | 6/11 | 6/11 | 6/11 | 6/11 | 6/11 | 6/11 | |

**IRS.V-ZERO penalty.** Finally, let us focus on $z_t^{\text{IRS.V-ZERO}}(\mathbf{a}_{1:t}, \omega) \triangleq r_t(\mathbf{a}_{1:t}, \omega) - \hat{\mu}_{a_t, n_{t-1}(\mathbf{a}_{1:t-1}, a_t)}$ under which the DM earns $\hat{\mu}_{a,n-1}(\omega)$ from the $n^{\text{th}}$ pull of arm $a$. Since the payoff from an arm changes over time as the Bayesian estimate evolves, playing only one arm is no longer optimal, unlike in the previous two cases. It can be easily verified that the optimal allocation is to play arm 1 six times and arm 2 two times, as visualized in the table below.

| | Payoffs under $z_t^{\text{IRS.V-ZERO}}$ | | | | | | | | Maximal payoff |
|---|---|---|---|---|---|---|---|---|---|
| | $n = 1$ | 2 | 3 | 4 | 5 | 6 | 7 | 8 | |
| Arm 1 | 3/4 | 3/5 | 4/6 | 5/7 | 6/8 | 6/9 | 6/10 | 6/11 | |
| Arm 2 | 1/2 | 2/3 | 2/4 | 2/5 | 3/6 | 4/7 | 5/8 | 6/9 | 5.314 |
| Arm 3 | 1/4 | 2/5 | 3/6 | 4/7 | 5/8 | 5/9 | 5/10 | 6/11 | |

**IRS.V-EMAX and the ideal penalty.** Regarding the penalty functions $z_t^{\text{IRS.V-EMAX}}$ and $z_t^{\text{ideal}}$, we cannot visualize the optimal solution with a table since the total payoff depends on the detailed sequence of pulls and not only the number of pulls. While omitting the visual proof of optimality, we have that the action sequence $\mathbf{a}_{1:8}^* = (1, 2, 2, 1, 1, 1, 1, 1)$ achieves the maximal payoff of 5.806 under $z_t^{\text{IRS.V-EMAX}}$, and $\mathbf{a}_{1:8}^* = (1, 1, 1, 1, 1, 1, 1, 1)$ achieves the maximal payoff of 6.063 under $z_t^{\text{ideal}}$. In particular for $z_t^{\text{ideal}}$, the maximal payoff depends only on the prior belief $\mathbf{y}$ and the time horizon $T$, irrespective of the outcome[3] $\omega$.

We have so far illustrated how the different penalty functions induce the different inner problems and the different best actions given the same outcome $\omega$. The readers may notice from the above examples that, as the penalty function becomes more complicated, the hindsight best action sequence becomes less dependent on a particular realization of $\omega$. Instead, it becomes more dependent on the prior belief.

## A.2 IRS Performance Bounds

The maximal payoffs above are calculated for a particular outcome given by Table 2. Recall that the IRS performance bound $W^z$ is defined as the expected value of the maximal payoff where the expectation is taken with respect to the randomness of outcome $\omega$ over its prior distribution $\mathcal{I}(T, \mathbf{y})$. We can obtain this value by simulation, i.e., by solving a bunch of inner problems with respect to the randomly generated outcomes $\omega^{(1)}, \omega^{(2)}, \ldots, \omega^{(S)}$ and taking the average of the maximal values. For this particular Beta-Bernoulli MAB setting ($T = 8$ with given priors), we obtain the following performance bounds:

| $W^0$ | $W^{\text{TS}}$ | $W^{\text{IRS.FH}}$ | $W^{\text{IRS.V-ZERO}}$ | $W^{\text{IRS.V-EMAX}}$ | $W^{\text{ideal}} = V^*$ |
|---|---|---|---|---|---|
| 6.805 | 6.429 | 6.279 | 6.111 | 6.075 | 6.063 |

We observe that the performance bounds are monotone, i.e., $W^0 > W^{\text{TS}} > W^{\text{IRS.FH}} > W^{\text{IRS.V-ZERO}} > W^{\text{IRS.V-EMAX}} > W^{\text{ideal}} = V^*$, which is consistent with Theorem 2.

## A.3 Illustration of the IRS Policy (IRS.V-Zero)

We illustrate how the policy $\pi^{\text{IRS.V-ZERO}}$ makes decisions sequentially when the true outcome $\omega$ is the one specified in Table 2. At $t = 1$, it first synthesizes a future scenario based on the prior belief (i.e.,

sampling $\tilde{\omega}_1 \sim \mathcal{I}(8, \mathbf{y}_0))$ and finds the best action sequence in the presence of penalties $z_t^{\text{IRS.V-ZERO}}$ in the belief that the sampled outcome $\tilde{\omega}_1$ is the ground truth. The following table shows an example in which $\pi^{\text{IRS.V-ZERO}}$ plays arm 1.

| $t = 1$ | Priors $\mathbf{y}_0$ | Payoffs with respect to $\tilde{\omega}_1 \sim \mathcal{I}(8, \mathbf{y}_0)$ | | | | | | | | Action |
|---|---|---|---|---|---|---|---|---|---|---|
| | | $n = 1$ | 2 | 3 | 4 | 5 | 6 | 7 | 8 | |
| Arm 1 | Beta$(3,1)$ | 3/4 | 4/5 | 5/6 | 6/7 | 7/8 | 7/9 | 8/10 | 9/11 | |
| Arm 2 | Beta$(1,1)$ | 1/2 | 1/3 | 1/4 | 1/5 | 1/6 | 1/7 | 2/8 | 3/9 | $a_1 = 1$ |
| Arm 3 | Beta$(1,3)$ | 1/4 | 1/5 | 1/6 | 1/7 | 1/8 | 1/9 | 1/10 | 2/11 | |

As a result of the first action ($a_1 = 1$), we observe that $R_{1,1} = 0$ (encoded in the true outcome $\omega$) and the associated belief is updated from Beta$(3,1)$ to Beta$(3,2)$ according to Bayes' rule. In order to make the next decision $a_2$ at time $t = 2$, $\pi^{\text{IRS.V-ZERO}}$ simulates an outcome for the remaining time horizon, i.e., $\tilde{\omega}_2 \sim \mathcal{I}(7, \mathbf{y}_1)$, independently of the outcome $\tilde{\omega}_1$ used at $t = 1$. Again, $\pi^{\text{IRS.V-ZERO}}$ finds the best action sequence for this new scenario and takes its first action.[4] The table below shows an instance of $\tilde{\omega}_2$ in which the policy will pull arm 2.

| $t = 2$ | Priors $\mathbf{y}_1$ | Payoffs with respect to $\tilde{\omega}_2 \sim \mathcal{I}(7, \mathbf{y}_1)$ | | | | | | | Action |
|---|---|---|---|---|---|---|---|---|---|
| | | $n = 1$ | 2 | 3 | 4 | 5 | 6 | 7 | |
| Arm 1 | Beta$(3,2)$ | 3/5 | 4/6 | 4/7 | 4/8 | 4/9 | 5/10 | 5/11 | |
| Arm 2 | Beta$(1,1)$ | 1/2 | 2/3 | 3/4 | 3/5 | 4/6 | 4/7 | 5/8 | $a_2 = 2$ |
| Arm 3 | Beta$(1,3)$ | 1/4 | 1/5 | 1/6 | 1/7 | 1/8 | 1/9 | 1/10 | |

We can update the prior of arm 2 as a new reward realization $R_{2,1} = 1$ is revealed. In the following decision epochs $t = 3, 4, \ldots$, the policy repeats the same decision-making procedure – (i) samples $\tilde{\omega}_t \sim \mathcal{I}(T - t + 1, \mathbf{y}_{t-1})$, (ii) solves the inner problem, and (iii) plays the best arm that the optimal solution suggests – while updating the priors as the true reward realizations are revealed sequentially.

The following table illustrates the last decision epoch. As there remains one time period only, the policy $\pi^{\text{IRS.V-ZERO}}$ tries to maximize $\hat{\mu}_{a,0}(\tilde{\omega}_7) = \bar{\mu}_a(\mathbf{y}_7)$, which is the expected mean reward given the prior at that moment. Such a decision is totally myopic, but it is Bayesian optimal.

| $t = 8$ | Priors $\mathbf{y}_7$ | Payoffs with respect to $\tilde{\omega}_7 \sim \mathcal{I}(1, \mathbf{y}_7)$ | Action |
|---|---|---|---|
| | | $n = 1$ | |
| Arm 1 | Beta$(6,3)$ | 6/9 | |
| Arm 2 | Beta$(2,2)$ | 2/4 | $a_8 = 1$ |
| Arm 3 | Beta$(1,3)$ | 1/4 | |

# B Algorithms in Detail

## B.1 Implementation of IRS.V-ZERO

We provide a pseudo-code of $\pi^{\text{IRS.V-ZERO}}$ introduced in §3.3. The same logic can be directly used to compute the performance bound $W^{\text{IRS.V-ZERO}}$ if the sampled outcome $\tilde{\omega}$ is replaced with the true outcome $\omega$.

---

**Algorithm 2:** IRS.V-ZERO policy

---
**Function** IRS.V-Zero$(T, \mathbf{y})$

1    $\tilde{\theta}_a \sim \mathcal{P}_a(y_a), \tilde{R}_{a,n} \sim \mathcal{R}_a(\tilde{\theta}), \forall n \in [T], \forall a \in [K]$
2    **for** $a = 1, \ldots, K$ **do**
3      $\tilde{y}_{a,0} \leftarrow y_a, \tilde{S}_{a,0} \leftarrow 0$
4      **for** $n = 1, \ldots, T$ **do**
5        $\tilde{S}_{a,n} \leftarrow \tilde{S}_{a,n-1} + \bar{\mu}_a(\tilde{y}_{a,n-1})$
6        $\tilde{y}_{a,n} \leftarrow \mathcal{U}_a(\tilde{y}_{a,n-1}, \tilde{R}_{a,n})$
     **end**
   **end**
7    $\tilde{M}_{0,0} \leftarrow 0, \tilde{M}_{0,n} \leftarrow -\infty, \forall n = 1, \ldots, T$
8    **for** $a = 1, \ldots, K$ **do**
9      **for** $n = 0, \ldots, T$ **do**
10       $\tilde{M}_{a,n} \leftarrow \max_{0 \le m \le n} \{\tilde{M}_{a-1,n-m} + \tilde{S}_{a,m}\}$
11       $\tilde{A}_{a,n} \leftarrow \operatorname{argmax}_{0 \le m \le n} \{\tilde{M}_{a-1,n-m} + \tilde{S}_{a,m}\}$
     **end**
   **end**
12    $m \leftarrow T$
13    **for** $a = K, \ldots, 1$ **do**
14      $\tilde{n}_a^* \leftarrow \tilde{A}_{a,m}$
15      $m \leftarrow m - \tilde{n}_a^*$
   **end**
16    **return** $\operatorname{argmax}_a \tilde{n}_a^*$

---

## B.2 IRS.V-EMAX

Under perfect information relaxation, the DM perfectly knows not only (i) what she will earn at future times but also (ii) how her belief will evolve as a result of her action sequence. The previous algorithms focus on the former component by making the DM to adjust the future rewards by conditioning (e.g., $\mathbb{E}[r_t(a_t)|\boldsymbol{\theta}]$, $\mathbb{E}[r_t(a_t)|\hat{\boldsymbol{\mu}}_{1:K,T-1}]$ and $\mathbb{E}[r_t(a_t)|\mathcal{F}_{t-1}]$). IRS.V-EMAX also focuses on the second component as well by charging an additional cost for using the information on her future belief transitions.

To motivate this in detail, recall that the ideal penalty $z_t^{\text{ideal}}$ (8) is

$$z_t^{\text{ideal}}(\mathbf{a}_{1:t}, \omega) \triangleq r_t(\mathbf{a}_{1:t}, \omega) - \mathbb{E}\left[r_t(\mathbf{a}_{1:t}, \omega)\,|\mathcal{F}_{t-1}(\mathbf{a}_{1:t-1}, \omega)\right] \tag{24}$$
$$+ V^*\left(T - t, \mathbf{y}_t(\mathbf{a}_{1:t}, \omega)\right) - \mathbb{E}\left[V^*\left(T - t, \mathbf{y}_t(\mathbf{a}_{1:t}, \omega)\right)|\mathcal{F}_{t-1}(\mathbf{a}_{1:t-1}, \omega)\right],$$

where $V^*(T - t, \mathbf{y}_t)$ measures the value of having a belief $\mathbf{y}_t$ at a future time $t + 1$. Note that, at the moment the DM takes an action $a_t$, her next belief state $\mathbf{y}_t = \mathcal{U}(\mathbf{y}_{t-1}, a_t, r_t)$ is not measurable with respect to the natural filtration $\mathcal{F}_{t-1}$ since the next observation $r_t$ is unknown. In DP terms, the conditional expectation $\mathbb{E}\left[V^*(T - t, \mathbf{y}_t)|\mathcal{F}_{t-1}\right]$ captures the expected value of (random) next state given the current state. Accordingly, the gap between its realized value and its expected value, $V^*(T - t, \mathbf{y}_t) - \mathbb{E}\left[V^*(T - t, \mathbf{y}_t)|\mathcal{F}_{t-1}\right]$, measures the additional gain from knowing the next belief state $\mathbf{y}_t$. In addition to the term $r_t - \mathbb{E}\left[r_t|\mathcal{F}_{t-1}\right] (= z_t^{\text{IRS.V-ZERO}})$, which measures the benefit from knowing which action will incur a large immediate reward, the ideal penalty also penalizes the long-term benefit from knowing which action will lead to a favorable belief state.

The penalty function $z_t^{\text{IRS.V-EMAX}}$ is obtained from $z_t^{\text{ideal}}$ by replacing $V^*(T, \mathbf{y})$ with $W^{\text{TS}}(T, \mathbf{y})$, which is rather tractable. The use of $W^{\text{TS}}(T, \mathbf{y}) \triangleq \mathbb{E}_{\boldsymbol{\theta} \sim \mathcal{P}(\mathbf{y})}\left[T \times \max_a \mu_a(\theta_a)\right]$ leads to a simple

expression for its conditional expectation: since $\boldsymbol{\theta}|\mathcal{F}_{t-1}$ is distributed with $\mathcal{P}(\mathbf{y}_{t-1})$, we have

$$\mathbb{E}\left[W^{\text{TS}}\left(T-t,\mathbf{y}_t\right)\middle|\mathcal{F}_{t-1}\right] = (T-t) \times \mathbb{E}\left[\max_a \mu_a(\theta_a)\middle|\mathcal{F}_{t-1}\right] \tag{25}$$

$$= (T-t) \times \mathbb{E}_{\boldsymbol{\theta}\sim\mathcal{P}(\mathbf{y}_{t-1})}\left[\max_a \mu_a(\theta_a)\right] \tag{26}$$

$$= W^{\text{TS}}\left(T-t,\mathbf{y}_{t-1}\right). \tag{27}$$

In the associated inner problem, the payoff that the DM earns at time $t$ is

$$r_t(\mathbf{a}_{1:t}) - z_t^{\text{IRS.V-EMAX}}(\mathbf{a}_{1:t}) \tag{28}$$

$$= \hat{\mu}_{a_t, n_{t-1}(\mathbf{a}_{1:t-1}, a_t)} - W^{\text{TS}}\left(T-t,\mathbf{y}_t(\mathbf{a}_{1:t})\right) + W^{\text{TS}}\left(T-t,\mathbf{y}_{t-1}(\mathbf{a}_{1:t-1})\right) \tag{29}$$

$$= \bar{\mu}_{a_t}(\mathbf{y}_{t-1}(\mathbf{a}_{1:t-1})) - W^{\text{TS}}\left(T-t,\mathbf{y}_t(\mathbf{a}_{1:t})\right) + W^{\text{TS}}\left(T-t,\mathbf{y}_{t-1}(\mathbf{a}_{1:t-1})\right), \tag{30}$$

which is completely determined by the prior belief $\mathbf{y}_{t-1}$ and the posterior belief $\mathbf{y}_t$.

We further observe that, given $\omega$, the future belief $\mathbf{y}_t(\mathbf{a}_{1:t}, \omega)$ depends only on how many times each arm has been pulled, irrespective of the sequence of the pulls. For example, consider two action sequences $\mathbf{a}_{1:t}^A = (1,1,2,1,2)$ and $\mathbf{a}_{1:t}^B = (2,1,1,2,1)$. Even though the order of observations would differ, in both cases the agent would observe $(R_{1,1}, R_{1,2}, R_{1,3})$ from arm 1 and $(R_{2,1}, R_{2,2})$ from arm 2 and end up with the same belief $\mathbf{y}_t(\mathbf{a}_{1:t}^A, \omega) = \mathbf{y}_t(\mathbf{a}_{1:t}^B, \omega)$. We may conclude from this observation that a belief state can be sufficiently parameterized with the pull counts $\mathbf{n}_{1:K} = (n_1, \ldots, n_K)$ instead of action sequence $\mathbf{a}_{1:t}$, that is, with $\mathbf{y}_t(\mathbf{n}_{1:K})$ instead of $\mathbf{y}_t(\mathbf{a}_{1:t})$.

Based on the observation above, we use the notation of $\mathbf{y}_t(\mathbf{n}_{1:K}, \omega)$ to denote the belief as a function of pull counts $\mathbf{n}_{1:K} \triangleq (n_1, \ldots, n_K) \in \mathbb{Z}^K$. Given the pull counts $\mathbf{n}_{1:K}$, we define the payoff of pulling an arm $a$ one more time after pulling each arm $n_1, \ldots, n_K$ times: with $t = \sum_{a=1}^K n_a$,

$$r^z(\mathbf{n}_{1:K}, a, \omega) \triangleq \bar{\mu}_a([\mathbf{y}_t(\mathbf{n}_{1:K}, \omega)]_a) - W^{\text{TS}}\left(T-t-1, \mathbf{y}_{t+1}(\mathbf{n}_{1:K} + \mathbf{e}_a, \omega)\right) \tag{31}$$
$$+ W^{\text{TS}}\left(T-t-1, \mathbf{y}_t(\mathbf{n}_{1:K}, \omega)\right)$$

where $\mathbf{e}_a \in \mathbb{N}_0^K$ is a basis vector such that $a^{\text{th}}$ component is one and the others are zero. Note that we used the fact that $\mathbb{E}\left[W^{\text{TS}}\left(T-t,\mathbf{y}_t\right)\middle|\mathcal{F}_{t-1}\right] = W^{\text{TS}}\left(T-t,\mathbf{y}_{t-1}\right)$.

Consider a subproblem of $(*)$ such that maximizes the total payoff given the number of pulls $\mathbf{n}_{1:K}$ across arm: with $t = \sum_{a=1}^K n_a$,

$$M(\mathbf{n}_{1:K}, \omega) \triangleq \max_{\mathbf{a}_{1:t} \in \mathcal{A}^t} \left\{ \sum_{s=1}^t r_s(\mathbf{a}_{1:s}, \omega) - z_s^{\text{IRS.V-EMAX}}(\mathbf{a}_{1:s}, \omega) \; : \; \sum_{s=1}^t \mathbf{1}\{a_s = a\} = n_a, \forall a \right\}. \tag{32}$$

Then, it should satisfy the Bellman equation

$$M(\mathbf{n}_{1:K}, \omega) = \max_{a \in \mathcal{A}: n_a \geq 1} \left\{ M(\mathbf{n}_{1:K} - \mathbf{e}_a, \omega) + r^z(\mathbf{n}_{1:K} - \mathbf{e}_a, a, \omega) \right\}. \tag{33}$$

For all feasible counts $\mathbf{n}_{1:K}$'s such that $\sum_{a=1}^K n_a \leq T$, we can compute $M(\mathbf{n}_{1:K}, \omega)$'s by sequentially solving (33) in an appropriate order. After all, we can obtain the maximal value to original inner problem $(*)$ by evaluating

$$\max_{\mathbf{n}_{1:K} \in N_T} \left\{ M(\mathbf{n}_{1:K}, \omega) \right\} \tag{34}$$

where $N_T \triangleq \{(n_1, \ldots, n_K) \in \mathbb{N}_0^K : \sum_{a=1}^K n_a = T\}$. The optimal action sequence $\mathbf{a}_{1:T}^*$ can be elicited by tracking $M(\mathbf{n}_{1:K}, \omega)$'s backward.

**Algorithm 3:** IRS.V-EMAX policy

---

**Function** `IRS.V-EMax`$(T, \mathbf{y})$

1    Sample an outcome $\tilde{\omega} \sim \mathcal{I}(T, \mathbf{y})$

2    $\tilde{y}_{a,0} \leftarrow y_a, \quad \tilde{y}_{a,n} \leftarrow \mathcal{U}_a(\tilde{y}_{a,n-1}, \tilde{R}_{a,n}), \quad \forall n \in [T], \quad \forall a \in [K]$

3    **for** *each* $\mathbf{n}_{1:K} \in N_{\leq T}$ **do**

4      $\tilde{\Gamma}[\mathbf{n}_{1:K}] \leftarrow \mathbb{E}_{\boldsymbol{\theta} \sim \mathcal{P}(\tilde{\mathbf{y}}(\mathbf{n}_{1:K}))} [\max_a \mu_a(\theta_a)]$

   **end**

5    **for** *each* $\mathbf{n}_{1:K} \in N_{<T}$ **do**

6      $\tilde{r}^z[\mathbf{n}_{1:K}, a] \leftarrow \bar{\mu}_a(\tilde{y}_{a,n_a-1}) + \left(T - \sum_{a=1}^{K} n_a - 1\right) \times \left(\tilde{\Gamma}[\mathbf{n}_{1:K}] - \tilde{\Gamma}[\mathbf{n}_{1:K} + \mathbf{e}_a]\right), \quad \forall a$

   **end**

7    $\tilde{M}[\mathbf{0}] \leftarrow 0$

8    **for** *each* $\mathbf{n}_{1:K} \in N_{\leq T} \setminus \{\mathbf{0}\}$ *in order* **do**

9      $\tilde{M}[\mathbf{n}_{1:K}] \leftarrow \max_{a:n_a>0} \left\{ \tilde{M}[\mathbf{n}_{1:K} - \mathbf{e}_a] + \tilde{r}^z[\mathbf{n}_{1:K} - \mathbf{e}_a, a] \right\}$

10      $\tilde{A}[\mathbf{n}_{1:K}] \leftarrow \operatorname{argmax}_{a:n_a>0} \left\{ \tilde{M}[\mathbf{n}_{1:K} - \mathbf{e}_a] + \tilde{r}^z[\mathbf{n}_{1:K} - \mathbf{e}_a, a] \right\}$

   **end**

11    $\mathbf{m}_{1:K} \leftarrow \operatorname{argmax}_{\mathbf{n}_{1:K} \in N_T} \left\{ \tilde{M}[\mathbf{n}_{1:K}] \right\}$

12    **for** $t = T, \ldots, 1$ **do**

13      $\tilde{a}_t^* \leftarrow \tilde{A}[\mathbf{m}_{1:K}]$

14      $m_{\tilde{a}_t^*} \leftarrow m_{\tilde{a}_t^*} - 1$

   **end**

15    **return** $\tilde{a}_1^*$

---

Here, $\tilde{\mathbf{y}}(\mathbf{n}_{1:K}) \triangleq (\tilde{y}_{1,n_1}, \ldots, \tilde{y}_{K,n_K})$, $N_{\leq T} \triangleq \{\mathbf{n}_{1:K} \in \mathbb{N}_0^K : \sum_a n_a \leq T\}$, $N_{<T} \triangleq \{\mathbf{n}_{1:K} \in \mathbb{N}_0^K : \sum_a n_a < T\}$, and in line 8, $\mathbf{n}_{1:K}$ iterates over $N_{\leq T} \setminus \{\mathbf{0}\}$ in an order that $\sum_{a=1}^{K} n_a$ is non-decreasing.

Computing all $M(\mathbf{n}_{1:K}, \omega)$'s requires $O(KT^K)$ operations since the number of possible belief states is limited to $|N_{\leq T}| = O(T^K)$, However, another practical issue is the cost of computing $W^{\text{TS}}(T, \mathbf{y}) = T \times \mathbb{E}_{\boldsymbol{\theta} \sim \mathcal{P}(\mathbf{y})}[\max_a \mu_a(\theta_a)]$ which has to be evaluated $O(T^K)$ times in total. There is no simple closed form expression in general, and it should be evaluated with numerical integration or sampling.

# C Proofs for §3

**Proposition 2** (Mean equivalence). *If the penalty function $z_t$ is dual feasible, it does not penalize any non-anticipating policy $\pi \in \Pi_{\mathbb{F}}$ in expectation, i.e.,*

$$\mathbb{E}_{\omega \sim \mathcal{I}(\mathbf{y})}^{\pi} \left[ \sum_{t=1}^{T} r_t(\mathbf{a}_{1:t}^{\pi}, \omega) - z_t(\mathbf{a}_{1:t}^{\pi}, \omega) \right] = \mathbb{E}_{\omega \sim \mathcal{I}(\mathbf{y})}^{\pi} \left[ \sum_{t=1}^{T} r_t(\mathbf{a}_{1:t}^{\pi}, \omega) \right] \equiv V(\pi, T, \mathbf{y}). \quad (35)$$

**Proof.** We define an appending operator $\oplus$ that concatenates an element into a vector such that $\mathbf{a}_{1:t} \equiv \mathbf{a}_{1:t-1} \oplus a_t$. When $\pi \in \Pi_{\mathbb{F}}$ and $z_t$ is dual feasible and $\omega$ is omitted for brevity, we have

$$\mathbb{E} \left[ \sum_{t=1}^{T} r_t(\mathbf{a}_{1:t}^{\pi}) - z_t(\mathbf{a}_{1:t}^{\pi}) \right] = \mathbb{E} \left[ \sum_{t=1}^{T} r_t(\mathbf{a}_{1:t}^{\pi}) - \mathbb{E}\left[ z_t(\mathbf{a}_{1:t}^{\pi}) \middle| \mathcal{F}_{t-1} \right] \right] \quad (36)$$

$$= \mathbb{E} \left[ \sum_{t=1}^{T} r_t(\mathbf{a}_{1:t}^{\pi}) - \mathbb{E}\left[ \sum_{a \in \mathcal{A}} z_t(\mathbf{a}_{1:t-1}^{\pi} \oplus a) \cdot \mathbf{1}\{a_t^{\pi} = a\} \middle| \mathcal{F}_{t-1} \right] \right] \quad (37)$$

$$= \mathbb{E} \left[ \sum_{t=1}^{T} \left( r_t(\mathbf{a}_{1:t}^{\pi}) - \sum_{a \in \mathcal{A}} \underbrace{\mathbb{E}\left[ z_t(\mathbf{a}_{1:t-1}^{\pi} \oplus a) \middle| \mathcal{F}_{t-1} \right]}_{=0} \cdot \mathbf{1}\{a_t^{\pi} = a\} \right) \right] \quad (38)$$

$$= \mathbb{E} \left[ \sum_{t=1}^{T} r_t(\mathbf{a}_{1:t}^{\pi}) \right]. \quad (39)$$

∎

## C.1 Proof of Theorem 1

**Weak duality.** Define $\mathcal{G}_t \triangleq \mathcal{F}_t \cup \sigma(\omega)$ and consider a relaxed policy space $\Pi_{\mathbb{G}} \triangleq \{\pi : a_t^{\pi} \text{ is } \mathcal{G}_{t-1}\text{-measurable}, \forall t\}$. Then, we have

$$V^*(T, \mathbf{y}) \triangleq \sup_{\pi \in \Pi_{\mathbb{F}}} \mathbb{E} \left[ \sum_{t=1}^{T} r_t(\mathbf{a}_{1:t}^{\pi}) \right] \overset{\text{Prop 2}}{=} \sup_{\pi \in \Pi_{\mathbb{F}}} \mathbb{E} \left[ \sum_{t=1}^{T} r_t(\mathbf{a}_{1:t}^{\pi}) - z_t(\mathbf{a}_{1:t}^{\pi}) \right] \quad (40)$$

$$\leq \sup_{\pi \in \Pi_{\mathbb{G}}} \mathbb{E} \left[ \sum_{t=1}^{T} r_t(\mathbf{a}_{1:t}^{\pi}) - z_t(\mathbf{a}_{1:t}^{\pi}) \right] = \mathbb{E} \left[ \max_{\mathbf{a}_{1:T} \in \mathcal{A}^T} \sum_{t=1}^{T} r_t(\mathbf{a}_{1:t}) - z_t(\mathbf{a}_{1:t}) \right] \quad (41)$$

$$= W^z(T, \mathbf{y}), \quad (42)$$

where the inequality holds since $\Pi_{\mathbb{F}} \subseteq \Pi_{\mathbb{G}}$. ∎

**Strong duality.** Fix $T$ and $\mathbf{y}$. Let $V_t^{\text{in}}(\mathbf{a}_{1:t-1}, \omega)$ and $Q_t^{\text{in}}(\mathbf{a}_{1:t-1}, a, \omega)$ be, respectively, the value function and the state-action value (Q-value) function that are associated with the inner problem (∗) given a particular outcome $\omega$ under the ideal penalty (8). With $V_{T+1}^{\text{in}} \equiv 0$, we have the Bellman equation for the inner problem:

$$Q_t^{\text{in}}(\mathbf{a}_{1:t-1}, a, \omega) \triangleq r_t(\mathbf{a}_{1:t-1} \oplus a, \omega) - z_t^{\text{ideal}}(\mathbf{a}_{1:t-1} \oplus a, \omega) + V_{t+1}^{\text{in}}(\mathbf{a}_{1:t-1} \oplus a, \omega), \quad (43)$$

$$V_t^{\text{in}}(\mathbf{a}_{1:t-1}, \omega) = \max_{a \in \mathcal{A}} \left\{ Q_t^{\text{in}}(\mathbf{a}_{1:t-1}, a, \omega) \right\}. \quad (44)$$

We argue by induction to show that

$$V_t^{\text{in}}(\mathbf{a}_{1:t-1}, \omega) = V^*(T - t + 1, \mathbf{y}_{t-1}(\mathbf{a}_{1:t-1}, \omega)), \quad (45)$$

$$Q_t^{\text{in}}(\mathbf{a}_{1:t-1}, a, \omega) = Q^*(T - t + 1, \mathbf{y}_{t-1}(\mathbf{a}_{1:t-1}, \omega), a), \quad (46)$$

for all $\mathbf{a}_{1:t-1} \in \mathcal{A}^{t-1}$, $a \in \mathcal{A}$ and $t \in [T+1]$.

As a terminal case, when $t = T + 1$, the claim holds trivially, since $V_{T+1}^{\text{in}}(\mathbf{a}_{1:T}, \omega) = 0 = V^*(0, \mathbf{y}_T(\mathbf{a}_{1:T}, \omega))$. Now assume that the claim holds for $t + 1$: $V_{t+1}^{\text{in}}(\mathbf{a}_{1:t}, \omega) = V^*(T - t, \mathbf{y}_t(\mathbf{a}_{1:t}, \omega))$ for all $\mathbf{a}_{1:t} \in \mathcal{A}^t$. For any $\mathbf{a}_{1:t-1} \in \mathcal{A}^{t-1}$ and $a \in \mathcal{A}$, then,

$$Q_t^{\text{in}}(\mathbf{a}_{1:t-1}, a, \omega) = r_t(\mathbf{a}_{1:t-1} \oplus a, \omega) - z_t^{\text{ideal}}(\mathbf{a}_{1:t-1} \oplus a, \omega) + V_{t+1}^{\text{in}}(\mathbf{a}_{1:t-1} \oplus a, \omega) \tag{47}$$

$$= \mathbb{E}\left[ r_t(\mathbf{a}_{1:t-1} \oplus a, \omega) + V^*(T - t, \mathbf{y}_t(\mathbf{a}_{1:t-1} \oplus a, \omega)) \middle| \mathcal{F}_{t-1}(\mathbf{a}_{1:t-1}, \omega) \right] \tag{48}$$

$$\underbrace{- V^*(T - t, \mathbf{y}_t(\mathbf{a}_{1:t-1} \oplus a, \omega)) + V_{t+1}^{\text{in}}(\mathbf{a}_{1:t-1} \oplus a, \omega)}_{=0} \tag{49}$$

$$= \mathbb{E}\left[ r_t(\mathbf{a}_{1:t-1} \oplus a, \omega) + V^*(T - t, \mathbf{y}_t(\mathbf{a}_{1:t-1} \oplus a, \omega)) \middle| \mathcal{F}_{t-1}(\mathbf{a}_{1:t-1}, \omega) \right] \tag{50}$$

$$= \mathbb{E}_{r \sim \mathcal{R}_a(\mathcal{P}_a([\mathbf{y}_{t-1}(\mathbf{a}_{1:t-1}, \omega)]_a))}\left[ r + V^*(T - t, \mathcal{U}(\mathbf{y}_{t-1}(\mathbf{a}_{1:t-1}, \omega), a, r)) \right] \tag{51}$$

$$= Q^*(T - t, \mathbf{y}_{t-1}(\mathbf{a}_{1:t-1}, \omega), a), \tag{52}$$

where the last equality follows from the original Bellman equation (3). Consequently,

$$V_t^{\text{in}}(\mathbf{a}_{1:t-1}, \omega) = \max_{a \in \mathcal{A}} \left\{ Q_t^{\text{in}}(\mathbf{a}_{1:t-1}, a, \omega) \right\} \tag{53}$$

$$= \max_{a \in \mathcal{A}} \left\{ Q^*(T - t, \mathbf{y}_{t-1}(\mathbf{a}_{1:t-1}, \omega), a) \right\} \tag{54}$$

$$= V^*(T - t, \mathbf{y}_{t-1}(\mathbf{a}_{1:t-1}, \omega)). \tag{55}$$

Therefore the claim holds for all $t = 1, \dots, T$. In particular for $t = 1$, we have

$$V_1^{\text{in}}(\emptyset, \omega) = V^*(T, \mathbf{y}), \quad Q_1^{\text{in}}(\emptyset, a, \omega) = Q^*(T, \mathbf{y}, a), \quad \forall \omega. \tag{56}$$

Note that the maximal value of the inner problem does not depend on $\omega$, which is deterministic with respect to the randomness of $\omega$. As its expected value, $W^{\text{ideal}}(T, \mathbf{y}) = V^*(T, \mathbf{y})$. ∎

## C.2   Proof of Remark 1

We proceed on the proof of strong duality. The policy $\pi^{\text{ideal}}$ solves the same inner problem with respect to a randomly sampled outcome $\tilde{\omega}$. When the remaining time is $T$ and the current belief is $\mathbf{y}$, it takes an action with the largest Q-value: together with (56), it yields

$$a^{\pi^{\text{ideal}}} = \operatorname*{argmax}_a Q_1^{\text{in}}(\emptyset, a, \tilde{\omega}) = \operatorname*{argmax}_a Q^*(T, \mathbf{y}, a). \tag{57}$$

Therefore, at each moment, no matter what $\tilde{\omega}$ is chosen, the policy $\pi^{\text{ideal}}$ always takes the same action that Bayesian optimal policy would take. Although there might be some ambiguity regarding tie-breaking in $\operatorname{argmax}$, it does not affect the expected performance. Therefore, $V(\pi^{\text{ideal}}, T, \mathbf{y}) = V^*(T, \mathbf{y})$. ∎

## C.3   Proof of Remark 2

First observe that $\mathbb{E}\left[ r_t(\mathbf{a}_{1:t}, \omega) \middle| \mathcal{F}_{t-1} \right] = \hat{\mu}_{a_t, n_{t-1}(\mathbf{a}_{1:t-1}, a_t)}(\omega)$. Also note that

$$\mathbb{E}\left[ \mathbb{E}\left( r_t(\mathbf{a}_{1:t}, \omega) \middle| \boldsymbol{\theta} \right) \middle| \mathcal{F}_{t-1} \right] = \mathbb{E}\left[ \mu_{a_t}(\theta_{a_t}) \middle| \mathcal{F}_{t-1} \right] = \hat{\mu}_{a_t, n_{t-1}(\mathbf{a}_{1:t-1}, a_t)}(\omega), \tag{58}$$

and

$$\mathbb{E}\left[ \mathbb{E}\left( r_t(\mathbf{a}_{1:t}, \omega) \middle| \hat{\boldsymbol{\mu}}_{1:K, T-1}(\omega) \right) \middle| \mathcal{F}_{t-1} \right] = \mathbb{E}\left[ \hat{\mu}_{a_t, T-1}(\omega) \middle| \mathcal{F}_{t-1} \right] = \hat{\mu}_{a_t, n_{t-1}(\mathbf{a}_{1:t-1}, a_t)}(\omega). \tag{59}$$

Therefore, $\mathbb{E}\left[ z_t^{\text{TS}} \middle| \mathcal{F}_{t-1} \right] = \mathbb{E}\left[ r_t \middle| \mathcal{F}_{t-1} \right] - \mathbb{E}\left[ \mathbb{E}(r_t | \boldsymbol{\theta}) \middle| \mathcal{F}_{t-1} \right] = 0$, and $\mathbb{E}\left[ z_t^{\text{IRS.FH}} \middle| \mathcal{F}_{t-1} \right] = \mathbb{E}\left[ r_t \middle| \mathcal{F}_{t-1} \right] - \mathbb{E}\left[ \mathbb{E}(r_t | \hat{\boldsymbol{\mu}}_{1:K, T-1}) \middle| \mathcal{F}_{t-1} \right] = 0$. The other penalty functions have a form of $z_t = X_t - \mathbb{E}[X_t | \mathcal{F}_{t-1}]$ for some $X_t$. Therefore, $\mathbb{E}\left[ z_t \middle| \mathcal{F}_{t-1} \right] = \mathbb{E}\left[ X_t - \mathbb{E}(X_t | \mathcal{F}_{t-1}) \middle| \mathcal{F}_{t-1} \right] = \mathbb{E}\left[ X_t - X_t \middle| \mathcal{F}_{t-1} \right] = 0$. ∎

# D   Proofs for §4

## D.1   Notes on Regularity

**Proposition 3.** *If $\mathbb{E}|R_{a,n}| < \infty$ for all $a$,*

$$\mathbb{E}|\mu_a(\theta_a)| < \infty \quad and \quad W^{\mathrm{TS}}(T, \mathbf{y}) < \infty. \tag{60}$$

**Proof.** By Jensen's inequality,

$$\mathbb{E}|\mu_a(\theta_a)| = \mathbb{E}\left[|\mathbb{E}(R_{a,n}|\theta_a)|\right] \leq \mathbb{E}\left[\mathbb{E}(|R_{a,n}||\theta_a)\right] = \mathbb{E}|R_{a,n}| < \infty. \tag{61}$$

Consequently,

$$\mathbb{E}\left[\max_a \mu_a(\theta_a)\right] \leq \mathbb{E}\left[\sum_{a=1}^{K} |\mu_a(\theta_a)|\right] = \sum_{a=1}^{K} \mathbb{E}|\mu_a(\theta_a)| < \infty. \tag{62}$$

$\blacksquare$

**Proposition 4.** *If $\mathbb{E}|R_{a,n}| < \infty$,*

$$\lim_{n \to \infty} \hat{\mu}_{a,n}(\omega) = \lim_{n \to \infty} \frac{1}{n} \sum_{i=1}^{n} R_{a,i} = \mu_a(\theta_a) \quad almost\ surely, \tag{63}$$

*where $\hat{\mu}_{a,n}(\omega) \triangleq \mathbb{E}[\mu_a(\theta_a)|R_{a,1}, \ldots, R_{a,n}]$.*

**Proof.** Fix $a$ and let $\mathcal{H}_n \triangleq \sigma(R_{a,1}, \ldots, R_{a,n})$. First note that, by the strong law of large numbers, $\lim_{n \to \infty} \frac{1}{n} \sum_{i=1}^{n} R_{a,i} = \mu_a(\theta_a)$ almost surely. Therefore, $\mu_a(\theta_a)$ is measurable with respect to $\mathcal{H}_\infty \triangleq \bigcup_n \mathcal{H}_n$. Also note that $\hat{\mu}_{a,n} = \mathbb{E}(\mu_a(\theta_a)|\mathcal{H}_n)$ is a Doob martingale adapted to $\mathcal{H}_n$. By Levy's upward theorem, since $\mu_a(\theta_a) \in \mathcal{L}^1$ by Proposition 3, $\hat{\mu}_{a,n}$ converges to $\mathbb{E}(\mu_a(\theta_a)|\mathcal{H}_\infty) = \mu_a(\theta_a)$ almost surely as $n \to \infty$. $\blacksquare$

## D.2   Proof of Proposition 1

**Asymptotic behavior of IRS.FH.** Let $\tilde{\omega}$ be the sampled outcome used by IRS.FH$(T, \mathbf{y})$. By Proposition 4, we have $\lim_{n \to \infty} \hat{\mu}_{a,n}(\tilde{\omega}) = \mu_a(\tilde{\theta}_a)$ for almost all $\tilde{\omega}$. This, together with the assumption that $\mu_i(\theta_i) \neq \mu_j(\theta_j)$ for $i \neq j$, since $\mathrm{argmax}_a \mu_a(\tilde{\theta}_a)$ is uniquely defined for almost all $\tilde{\omega}$, yields

$$\mathrm{argmax}_a \mu_a(\tilde{\theta}_a) = \mathrm{argmax}_a \lim_{n \to \infty} \hat{\mu}_{a,n}(\tilde{\omega}) = \lim_{n \to \infty} \mathrm{argmax}_a \hat{\mu}_{a,n}(\tilde{\omega}) \quad \text{a.s.} \tag{64}$$

Since almost-sure convergence guarantees convergence in distribution, for any $a$,

$$\lim_{T \to \infty} \mathbb{P}\left[\mathrm{IRS.FH}(T, \mathbf{y}) = a\right] = \lim_{T \to \infty} \mathbb{P}\left[\mathrm{argmax}_{a'} \hat{\mu}_{a',T-1}(\tilde{\omega}) = a\right] \tag{65}$$

$$= \mathbb{P}\left[\mathrm{argmax}_{a'} \mu_{a'}(\tilde{\theta}_{a'}) = a\right] \tag{66}$$

$$= \mathbb{P}\left[\mathrm{TS}(\mathbf{y}) = a\right]. \tag{67}$$

Note that we are not particularly assuming that IRS.FH$(T, \mathbf{y})$ and TS$(\mathbf{y})$ share the randomness. The sampled parameters used in TS$(\mathbf{y})$ are not necessarily the ones used in IRS.FH$(T, \mathbf{y})$, but their distributions are identical since they are drawn from the same prior. $\blacksquare$

**Asymptotic behavior of IRS.V-ZERO.** Let $a_T^\circ(\tilde{\omega}) \triangleq$ IRS.V-ZERO$(T, \mathbf{y})$ in which $\tilde{\omega}$ is used, and let $a^{\mathrm{TS}}(\tilde{\omega}) \triangleq \mathrm{argmax}_a \mu_a(\tilde{\theta}_a)$. As above, it suffices to show that $\lim_{T \to \infty} a_T^\circ(\tilde{\omega}) = a^{\mathrm{TS}}(\tilde{\omega})$ for almost all $\tilde{\omega}$. We now fix $\tilde{\omega}$ and omit it from the proof for brevity.

Define

$$\Delta \triangleq \min_{a \neq a^{\mathrm{TS}}} \left|\mu_{a^{\mathrm{TS}}}(\tilde{\theta}_{a^{\mathrm{TS}}}) - \mu_a(\tilde{\theta}_a)\right| \quad and \quad M \triangleq \sup_{a \in \mathcal{A}, n \geq 0} |\hat{\mu}_{a,n}|. \tag{68}$$

We have $0 < \Delta < 2M < \infty$ almost surely since $\mu_i(\tilde{\theta}_i) \neq \mu_j(\tilde{\theta}_j)$ for $i \neq j$ and $\lim_{n \to \infty} \hat{\mu}_{a,n} = \mu_a(\tilde{\theta}_a) < \infty$ almost surely for all $a$. In addition, there exists $N \in \mathbb{N}$ such that

$$\left| \hat{\mu}_{a,n} - \mu_a(\tilde{\theta}_a) \right| < \frac{\Delta}{4}, \quad \forall n \geq N, \quad \forall a \in \mathcal{A}. \tag{69}$$

For such $N$, we have

$$\inf_{n \geq N} \hat{\mu}_{a^{\text{TS}},n} \geq \sup_{n \geq N} \hat{\mu}_{a,n} + \frac{\Delta}{2}, \quad \forall a \neq a^{\text{TS}}. \tag{70}$$

Note that $a^{\text{TS}}$, $\Delta$, $M$, and $N$ are determined only by $\tilde{\omega}$, independently of $T$.

To argue by contradiction, suppose that $a_T^{\circ} \neq a^{\text{TS}}$ for some large $T$ such that $T \geq 2N + \frac{8MN}{\Delta} + 2$. Define the optimal allocation to the inner problem of IRS.V-ZERO for such $T$:

$$\mathbf{n}_{1:K}^{\circ} \triangleq \underset{\mathbf{n}_{1:K} \in N_T}{\operatorname{argmax}} \left\{ \sum_{a=1}^{K} \sum_{s=1}^{n_a} \hat{\mu}_{a,s-1} \right\}, \tag{71}$$

where the ties are broken arbitrarily in $\operatorname{argmax}\{\}$. Policy $\pi^{\text{IRS.V-ZERO}}$'s selection rule, $a_T^{\circ} = \operatorname{argmax}_a n^{\circ}(a)$, implies that $n^{\circ}(a_T^{\circ}) \geq \lfloor \frac{T}{2} \rfloor \, (> N)$.

**Case 1:** If $n^{\circ}(a^{\text{TS}}) \geq N$, consider a deviation of $\mathbf{n}_{1:K}^{\circ}$ that plays $a^{\text{TS}}$ one more time but plays $a_T^{\circ}$ one less time: define $\mathbf{n}_{1:K}^{\dagger}$ such that $n^{\dagger}(a^{\text{TS}}) = n^{\circ}(a^{\text{TS}}) + 1$, $n^{\dagger}(a_T^{\circ}) = n^{\circ}(a_T^{\circ}) - 1$ and $n^{\dagger}(a) = n^{\circ}(a)$ for $a \notin \{a^{\text{TS}}, a_T^{\circ}\}$. Then, since $n^{\circ}(a^{\text{TS}}) \geq N$ and $n^{\circ}(a_T^{\circ}) \geq N$,

$$\sum_{a=1}^{K} \sum_{s=1}^{n^{\dagger}(a)} \hat{\mu}_{a,s-1} - \sum_{a=1}^{K} \sum_{s=1}^{n^{\circ}(a)} \hat{\mu}_{a,s-1} = \hat{\mu}_{a^{\text{TS}},n^{\circ}(a^{\text{TS}})} - \hat{\mu}_{a_T^{\circ},n^{\circ}(a_T^{\circ})-1} \geq \frac{\Delta}{2} > 0, \tag{72}$$

where the inequality follows from (70). The allocation $\mathbf{n}_{1:K}^{\dagger}$ achieves a strictly better payoff than $\mathbf{n}_{1:K}^{\circ}$, which contradicts the assumption that $\mathbf{n}_{1:K}^{\circ}$ is an optimal allocation.

**Case 2:** If $n^{\circ}(a^{\text{TS}}) < N$, consider a deviation $\mathbf{n}_{1:K}^{\dagger}$ such that

$$n^{\dagger}(a) \triangleq \begin{cases} n^{\circ}(a^{\text{TS}}) + (n^{\circ}(a_T^{\circ}) - N) & \text{if } a = a^{\text{TS}}, \\ N & \text{if } a = a_T^{\circ}, \\ n^{\circ}(a) & \text{otherwise.} \end{cases} \tag{73}$$

By making this allocation, we have

$$\sum_{a=1}^{K} \sum_{s=1}^{n^{\dagger}(a)} \hat{\mu}_{a,s-1} - \sum_{a=1}^{K} \sum_{s=1}^{n^{\circ}(a)} \hat{\mu}_{a,s-1} \tag{74}$$

$$= \sum_{s=n^{\circ}(a^{\text{TS}})+1}^{n^{\circ}(a^{\text{TS}})+(n^{\circ}(a_T^{\circ})-N)} \hat{\mu}_{a^{\text{TS}},s-1} - \sum_{s=N+1}^{n^{\circ}(a_T^{\circ})} \hat{\mu}_{a_T^{\circ},s-1} \tag{75}$$

$$\geq -(N - n^{\circ}(a^{\text{TS}})) \cdot 2M + \sum_{s=N+1}^{n^{\circ}(a_T^{\circ})} \hat{\mu}_{a^{\text{TS}},s-1} - \sum_{s=N+1}^{n^{\circ}(a_T^{\circ})} \hat{\mu}_{a_T^{\circ},s-1} \tag{76}$$

$$\geq -(N - n^{\circ}(a^{\text{TS}})) \cdot 2M + (n^{\circ}(a_T^{\circ}) - N) \cdot \frac{\Delta}{2} \tag{77}$$

$$\geq (n^{\circ}(a_T^{\circ}) - N) \cdot \frac{\Delta}{2} - 2NM. \tag{78}$$

Since $T \geq 2N + \frac{8MN}{\Delta} + 2$ and $n^{\circ}(a_T^{\circ}) \geq \lfloor \frac{T}{2} \rfloor$, the last term is strictly positive, which means that $\mathbf{n}_{1:K}^{\dagger}$ is strictly better than $\mathbf{n}_{1:K}^{\circ}$, a contradiction.

We've shown that for almost all $\tilde{\omega}$, when $T$ is large enough, the optimal allocation $\mathbf{n}_{1:K}^{\circ}$ must allocate more than a half of the pulls on the arm $a^{\text{TS}} = \operatorname{argmax}_a \mu_a(\tilde{\theta}_a)$. Therefore, $\lim_{T \to \infty} a_T^{\circ}(\tilde{\omega}) = a^{\text{TS}}(\tilde{\omega})$ for almost all $\tilde{\omega}$, which completes the proof.

## D.3 Proof of Theorem 2

### D.3.1 Proof of "$W^{\text{TS}}(T,\mathbf{y}) \geq W^{\text{IRS.FH}}(T,\mathbf{y})$"

**Proof.** It immediately follows from Jensen's inequality: since $\max(\ldots)$ is a convex function,

$$W^{\text{TS}}(T,\mathbf{y}) = T \times \mathbb{E}\left[\max_a \mu_a(\theta_a)\right] \geq T \times \mathbb{E}\left[\max_a \mathbb{E}\left(\mu_a(\theta_a)|\,\hat{\boldsymbol{\mu}}_{1:K,T-1}\right)\right] = W^{\text{IRS.FH}}(T,\mathbf{y}). \quad (79)$$

∎

### D.3.2 Proof of "$W^{\text{IRS.FH}}(T,\mathbf{y}) \geq W^{\text{IRS.V-ZERO}}(T,\mathbf{y})$"

**Lemma 1** (Variant of Jensen's inequality). *Suppose that $\varphi : \mathbb{R} \mapsto \mathbb{R}$ is an **increasing** (deterministic) function. Then, for any real-valued random variable $X$ such that $\mathbb{E}|X| < \infty$,*

$$\mathbb{E}\left[\max\left\{X + \varphi(X), 0\right\}\right] \geq \mathbb{E}\left[\max\left\{\mathbb{E}(X) + \varphi(X), 0\right\}\right]. \quad (80)$$

**Proof.** Define $\mu \triangleq \mathbb{E}(X)$ and $f_x(t) \triangleq \max\{t + \varphi(x), 0\}$. Since $f_x(\cdot)$ is a convex function for each $x \in \mathbb{R}$,

$$f_x(t) \geq f_x(\mu) + (t - \mu) \cdot f_x'(\mu) = \max\{\mu + \varphi(x), 0\} + (t - \mu) \cdot \mathbf{1}\{\mu + \varphi(x) \geq 0\}, \quad \forall t, \quad \forall x. \quad (81)$$

By setting $t = x$, we get

$$\max\{x + \varphi(x), 0\} = f_x(x) \geq \max\{\mu + \varphi(x), 0\} + (x - \mu) \cdot \mathbf{1}\{\mu + \varphi(x) \geq 0\}, \quad \forall x. \quad (82)$$

Note that, since $\mathbf{1}\{\mu + \varphi(x) \geq 0\}$ is increasing in $x$, (i) for any $x \geq \mu$, $(x - \mu) \geq 0$ and $\mathbf{1}\{\mu + \varphi(x)\} \geq \mathbf{1}\{\mu + \varphi(\mu)\}$, and (ii) for any $x < \mu$, $(x - \mu) < 0$ and $\mathbf{1}\{\mu + \varphi(x)\} \leq \mathbf{1}\{\mu + \varphi(\mu)\}$. Therefore,

$$(x - \mu) \cdot \mathbf{1}\{\mu + \varphi(x) \geq 0\} \geq (x - \mu) \cdot \mathbf{1}\{\mu + \varphi(\mu) \geq 0\}, \quad \forall x \in \mathbb{R}. \quad (83)$$

Combining this with (82), we get

$$\max\{x + \varphi(x), 0\} \geq \max\{\mu + \varphi(x), 0\} + (x - \mu) \cdot \mathbf{1}\{\mu + \varphi(\mu) \geq 0\}, \quad \forall x \in \mathbb{R}. \quad (84)$$

For random variable $X$, by taking expectation, we get

$$\mathbb{E}\left[\max\{X + \varphi(X), 0\}\right] \geq \mathbb{E}\left[\max\{\mu + \varphi(X), 0\} + (X - \mu) \cdot \mathbf{1}\{\mu + \varphi(\mu) \geq 0\}\right] \quad (85)$$
$$\geq \mathbb{E}\left[\max\{\mu + \varphi(X), 0\}\right] + \mathbb{E}(X - \mu) \cdot \mathbf{1}\{\mu + \varphi(\mu) \geq 0\} \quad (86)$$
$$= \mathbb{E}\left[\max\{\mu + \varphi(X), 0\}\right]. \quad (87)$$

∎

**Corollary 1.** *On a probability space $(\Omega, \mathcal{F}, \mathbb{P})$, let $\varphi(x, \omega) : \mathbb{R} \times \Omega \mapsto \mathbb{R}$ be a function such that (i) the mapping $x \mapsto \varphi(x, \omega)$ is **increasing** for each $\omega \in \Omega$ and (ii) for some sub-$\sigma$-field $\mathcal{H} \subseteq \mathcal{F}$, the mapping $\omega \mapsto \varphi(x, \omega)$ is $\mathcal{H}$-**measurable** for each $x \in \mathbb{R}$ (i.e., $\varphi(\cdot, \omega)$ is a deterministic function conditioned on $\mathcal{H}$). Then*

$$\mathbb{E}\left[\max\left\{X(\omega) + \varphi(X(\omega), \omega), 0\right\}\right] \geq \mathbb{E}\left[\max\left\{\mathbb{E}(X|\mathcal{H})(\omega) + \varphi(X(\omega), \omega), 0\right\}\right]. \quad (88)$$

**Proof.** Define

$$\mu(\omega) \triangleq \mathbb{E}(X|\mathcal{H})(\omega), \quad I(\omega) \triangleq \mathbf{1}\{\mu(\omega) + \varphi(\mu(\omega), \omega) \geq 0\}. \quad (89)$$

By (84), we have

$$\max\{x + \varphi(x, \omega), 0\} \geq \max\{\mu(\omega) + \varphi(x, \omega), 0\} + (x - \mu(\omega)) \cdot I(\omega), \quad \forall x \in \mathbb{R}, \quad \text{for each } \omega \in \Omega. \quad (90)$$

Since $\mu(\omega)$ and $I(\omega)$ are $\mathcal{H}$-measurable,

$$\mathbb{E}\left[\max\{X(\omega) + \varphi(X(\omega), \omega), 0\}\right] \geq \mathbb{E}\left[\max\{\mu(\omega) + \varphi(X(\omega), \omega), 0\} + (X(\omega) - \mu(\omega)) \cdot I(\omega)\right] \quad (91)$$
$$= \mathbb{E}\left[\mathbb{E}\left(\max\{\mu(\omega) + \varphi(X(\omega), \omega), 0\} + (X(\omega) - \mu(\omega)) \cdot I(\omega)|\,\mathcal{H}\right)\right] \quad (92)$$
$$= \mathbb{E}\left[\max\{\mu(\omega) + \varphi(X(\omega), \omega), 0\}\right] + \mathbb{E}\left[\mathbb{E}\left((X(\omega) - \mu(\omega)) \cdot I(\omega)|\,\mathcal{H}\right)\right] \quad (93)$$
$$= \mathbb{E}\left[\max\{\mathbb{E}(X|\mathcal{H})(\omega) + \varphi(X(\omega), \omega), 0\}\right] \quad (94)$$
$$+ \mathbb{E}\left[\underbrace{\left(\mathbb{E}(X|\mathcal{H})(\omega) - \mu(\omega)\right)}_{=0} \cdot I(\omega)\right] \quad (95)$$
$$= \mathbb{E}\left[\max\{\mathbb{E}(X|\mathcal{H})(\omega) + \varphi(X(\omega), \omega), 0\}\right]. \quad (96)$$

∎

**Corollary 2.** *On a probability space $(\Omega, \mathcal{F}, \mathbb{P})$, let $(C_0, \ldots, C_T)$ be $\mathcal{H}$-measurable real-valued random variables for some sub-$\sigma$-field $\mathcal{H} \subseteq \mathcal{F}$ (i.e., $C_i$'s are constants conditioned on $\mathcal{H}$). Then*

$$\mathbb{E}\left[\max_{0 \leq i \leq T} \left\{(i-n)^+ \times X + C_i\right\}\right] \geq \mathbb{E}\left[\max_{0 \leq i \leq T} \left\{\mathbb{E}\left(X \mid \mathcal{H}\right) \cdot \mathbf{1}\{i \geq n+1\} + (i-n-1)^+ \times X + C_i\right\}\right] \tag{97}$$

*for any $n = 0, 1, \ldots, T$.*

**Proof.** When $n = T$, both sides become $\mathbb{E}\left[\max_{0 \leq i \leq T}\{C_i\}\right]$, which makes the claim true. Fix $n < T$ and define

$$\varphi(x, \omega) \triangleq \max_{n+1 \leq i \leq T} \left\{(i-n-1) \times x + C_i(\omega)\right\} - \max_{0 \leq i \leq n}\{C_i(\omega)\}. \tag{98}$$

Note that $\varphi(x, \omega)$ satisfies the conditions in Corollary 1. By Corollary 1,

$$\mathbb{E}\left[\max_{0 \leq i \leq T}\left\{(i-n)^+ \times X + C_i\right\}\right] \tag{99}$$

$$= \mathbb{E}\left[\max\left\{\max_{n+1 \leq i \leq T}\left\{(i-n) \times X + C_i\right\}, \quad \max_{0 \leq i \leq n} C_i\right\}\right] \tag{100}$$

$$= \mathbb{E}\left[\max\left\{X + \max_{n+1 \leq i \leq T}\left\{(i-n-1) \times X + C_i\right\}, \quad \max_{0 \leq i \leq n} C_i\right\}\right] \tag{101}$$

$$= \mathbb{E}\left[\max\left\{\underbrace{X(\omega) + \max_{n+1 \leq i \leq T}\left\{(i-n-1) \times X(\omega) + C_i(\omega)\right\} - \max_{0 \leq i \leq n} C_i(\omega)}_{=\varphi(X(\omega), \omega)}, \quad 0\right\} + \max_{0 \leq i \leq n} C_i(\omega)\right] \tag{102}$$

$$\geq \mathbb{E}\left[\max\left\{\mathbb{E}\left(X \mid \mathcal{H}\right)(\omega) + \max_{n+1 \leq i \leq T}\left\{(i-n-1) \times X(\omega) + C_i(\omega)\right\} - \max_{0 \leq i \leq n} C_i(\omega), \quad 0\right\} + \max_{0 \leq i \leq n} C_i(\omega)\right] \tag{103}$$

$$= \mathbb{E}\left[\max\left\{\max_{n+1 \leq i \leq T}\left\{\mathbb{E}\left(X \mid \mathcal{H}\right) + (i-n-1) \times X + C_i\right\}, \quad \max_{0 \leq i \leq n} C_i\right\}\right] \tag{104}$$

$$= \mathbb{E}\left[\max_{0 \leq i \leq T}\left\{\mathbb{E}\left(X \mid \mathcal{H}\right) \cdot \mathbf{1}\{i \geq n+1\} + (i-n-1)^+ \times X + C_i\right\}\right]. \tag{105}$$

∎

**Proof of "$W^{\text{IRS.FH}}(T, \mathbf{y}) \geq W^{\text{IRS.V-ZERO}}(T, \mathbf{y})$."** Define

$$N_T \triangleq \left\{\mathbf{n}_{1:K} \in \mathbb{N}_0^K : \sum_{a=1}^{K} n_a = T\right\} \quad \text{and} \quad S_a(n_a) \triangleq \sum_{i=1}^{n_a} \hat{\mu}_{a, i-1}. \tag{106}$$

What we want to show is

$$W^{\text{IRS.FH}} \equiv \mathbb{E}\left[T \times \max_a\{\hat{\mu}_{a, T-1}\}\right] = \mathbb{E}\left[\max_{\mathbf{n}_{1:K} \in N_T}\left\{\sum_{a=1}^{K} n_a \times \hat{\mu}_{a, T-1}\right\}\right] \tag{107}$$

$$\geq \mathbb{E}\left[\max_{\mathbf{n}_{1:K} \in N_T}\left\{\sum_{a=1}^{K} S_a(n_a)\right\}\right] \equiv W^{\text{IRS.V-ZERO}}. \tag{108}$$

Further define

$$U_{k,n} \triangleq \mathbb{E}\left[\max_{\mathbf{n}_{1:K} \in N_T}\left\{\left(\sum_{a=1}^{k-1} S_a(n_a)\right) + \left(S_k(n_k \wedge n) + (n_k - n)^+ \times \hat{\mu}_{a, T-1}\right) + \left(\sum_{a=k+1}^{K} n_a \times \hat{\mu}_{a, T-1}\right)\right\}\right], \tag{109}$$

where $a \wedge b \triangleq \min(a, b)$. Observe that $W^{\text{IRS.FH}} = U_{1,0}$, $W^{\text{IRS.V-ZERO}} = U_{K,T}$, and $U_{k+1,0} = U_{k,T}$. Therefore, it suffices to show that

$$U_{k,n} \geq U_{k,n+1}, \quad \forall k = 1, \ldots, K, \quad \forall n = 0, \ldots, T-1. \tag{110}$$

Fix $k$ and $n$. Define a sub-$\sigma$-field

$$\mathcal{H} \triangleq \sigma\left(\{R_{a,s}\}_{a=k, 1 \leq s \leq n} \cup \{R_{a,s}\}_{a \neq k, 1 \leq s \leq T-1}\right). \tag{111}$$

For each $i = 0, \ldots, T$, define

$$C_i \triangleq \max\left\{ \left(\sum_{a=1}^{k-1} S_a(n_a)\right) + S_k(i \wedge n) + \left(\sum_{a=k+1}^{K} n_a \times \hat{\mu}_{a,T-1}\right) \; : \; \mathbf{n}_{1:K} \in N_T, n_k = i \right\}. \quad (112)$$

Note that $C_i$'s are $\mathcal{H}$-measurable and

$$U_{k,n} = \mathbb{E}\left[ \max_{0 \le i \le T} \left\{ (i-n)^+ \times \hat{\mu}_{k,T-1} + C_i \right\} \right]. \quad (113)$$

With $X \triangleq \hat{\mu}_{a,T-1}$,

$$U_{k,n} = \mathbb{E}\left[ \max_{0 \le i \le T} \left\{ (i-n)^+ \times X + C_i \right\} \right] \quad (114)$$

$$\overset{\text{Corollary 2}}{\ge} \mathbb{E}\left[ \max_{0 \le i \le T} \left\{ \mathbb{E}\left(X|\mathcal{H}\right) \cdot \mathbf{1}\{i \ge n+1\} + (i-n-1)^+ \times X + C_i \right\} \right] \quad (115)$$

$$\overset{\text{(a)}}{=} \mathbb{E}\left[ \max_{0 \le i \le T} \left\{ \hat{\mu}_{k,n} \cdot \mathbf{1}\{i \ge n+1\} + (i-n-1)^+ \times \hat{\mu}_{a,T-1} + C_i \right\} \right] \quad (116)$$

$$\overset{\text{(b)}}{=} U_{k,n+1}. \quad (117)$$

Equation (a) holds since $\mathbb{E}\left(X|\mathcal{H}\right) = \mathbb{E}\left(\hat{\mu}_{k,T-1}|\mathcal{H}\right) = \mathbb{E}\left(\hat{\mu}_{k,T-1}|R_{k,1}, \ldots, R_{k,n}\right) = \hat{\mu}_{a,n}$, and equation (b) holds since $S_k(i \wedge n) + \hat{\mu}_{k,n} \cdot \mathbf{1}\{i \ge n+1\} = \sum_{s=1}^{n} \hat{\mu}_{k,s-1} \cdot \mathbf{1}\{i \ge s\} + \hat{\mu}_{k,n} \cdot \mathbf{1}\{i \ge n+1\} = \sum_{s=1}^{n+1} \hat{\mu}_{k,s-1} \cdot \mathbf{1}\{i \ge s\} = S_k(i \wedge (n+1))$. ∎

## D.4 Proof of Theorem 3

As in §C.1, we define the Q-values of the inner problem given a particular outcome $\omega$, a penalty function $z_t(\cdot)$, a time horizon $T$, and a prior belief $\mathbf{y}$.

$$Q_t^{z,\text{in}}(\mathbf{a}_{1:t-1}, a, \omega; T, \mathbf{y}) = r_t(\mathbf{a}_{1:t-1} \oplus a, \omega) - z_t(\mathbf{a}_{1:t-1} \oplus a, \omega; T, \mathbf{y}) \quad (118)$$
$$+ V_{t+1}^{z,\text{in}}(\mathbf{a}_{1:t-1} \oplus a, \omega; T, \mathbf{y}),$$

$$V_t^{z,\text{in}}(\mathbf{a}_{1:t-1}, \omega; T, \mathbf{y}) = \max_{a \in \mathcal{A}} \left\{ Q_t^{z,\text{in}}(\mathbf{a}_{1:t-1}, a, \omega; T, \mathbf{y}) \right\}, \quad (119)$$

with $V_{T+1}^{z,\text{in}}(\cdot, \omega; T, \mathbf{y}) \equiv 0$. Additionally define the total payoff of an action sequence and the hindsight best action under penalties:

$$\mathcal{S}^z(\mathbf{a}_{1:T}, \omega; T, \mathbf{y}) \triangleq \sum_{t=1}^{T} r_t(\mathbf{a}_{1:t}, \omega) - z_t(\mathbf{a}_{1:t}, \omega; T, \mathbf{y}), \quad (120)$$

$$a_t^{z,*}(\mathbf{a}_{1:t-1}, \omega; T, \mathbf{y}) \triangleq \operatorname*{argmax}_{a \in \mathcal{A}} \left\{ Q_t^{z,\text{in}}(\mathbf{a}_{1:t-1}, a, \omega; T, \mathbf{y}) \right\}. \quad (121)$$

We have $V_1^{z,\text{in}}(\emptyset, \omega; T, \mathbf{y}) = \max_{\mathbf{a}_{1:T} \in \mathcal{A}^T} \mathcal{S}^z(\mathbf{a}_{1:T}, \omega; T, \mathbf{y})$.

**Proposition 5** (Suboptimality decomposition). *Given a non-anticipating policy $\pi \in \Pi_{\mathbb{F}}$ and a dual-feasible penalty function $z_t$,*

$$W^z(T, \mathbf{y}) - V(\pi, T, \mathbf{y}) = \mathbb{E}\left[ \max_{\mathbf{a}_{1:T}} \left\{ \mathcal{S}^z(\mathbf{a}_{1:T}, \omega; T, \mathbf{y}) \right\} - \mathcal{S}^z(\mathbf{a}_{1:T}^\pi, \omega; T, \mathbf{y}) \right] \quad (122)$$

$$= \mathbb{E}\left[ \sum_{t=1}^{T} \max_{a} \left\{ Q_t^{z,\text{in}}(\mathbf{a}_{1:t-1}^\pi, a, \omega; T, \mathbf{y}) \right\} - Q_t^{z,\text{in}}(\mathbf{a}_{1:t-1}^\pi, a_t^\pi, \omega; T, \mathbf{y}) \right], \quad (123)$$

*where the expectation is taken with respect to the randomness of outcome $\omega$ and the randomness of policy $\pi$.*

**Proof.** The first equality immediately follows from the definition of $W^z$ and mean equivalence (Proposition 2). Now fix $\omega$, $T$, and $\mathbf{y}$. Consider the (pathwise) suboptimality of the action sequence

$\mathbf{a}_{1:T}^{\pi}$ compared to the clairvoyant optimal solution. It can be decomposed into the instantaneous suboptimalty incurred by the individual action at each time:

$$\max_{\mathbf{a}_{1:T}} \left\{ \mathcal{S}^z(\mathbf{a}_{1:T}) \right\} - \mathcal{S}^z(\mathbf{a}_{1:T}^{\pi}) = \sum_{t=1}^{T} \max_a \left\{ Q_t^{z,\mathrm{in}}(\mathbf{a}_{1:t-1}, a) \right\} - Q_t^{z,\mathrm{in}}(\mathbf{a}_{1:t-1}^{\pi}, a_t^{\pi}). \qquad (124)$$

By taking expectation, we obtain the second equality. ∎

Define a shift operator $\mathcal{M}_t : \mathcal{A}^t \times \Omega \mapsto \Omega$,

$$\mathcal{M}_t(\mathbf{a}_{1:t}, \omega) \triangleq \left( R_{a,n_a} : \forall n_a > n_t(\mathbf{a}_{1:t}, a), \forall a \in \mathcal{A} \right). \qquad (125)$$

The shifted outcome $\mathcal{M}_{t-1}(\mathbf{a}_{1:t-1}, \omega)$ encodes the remaining reward realizations after taking $\mathbf{a}_{1:t-1}$.

**Remark 4** (Recursive structure of remaining uncertainties). *Conditioned on $\mathcal{F}_{t-1}(\mathbf{a}_{1:t-1}, \omega; T, \mathbf{y})$, the remaining uncertainties are sufficiently described by $\mathbf{y}_{t-1}(\mathbf{a}_{1:t-1}, \omega; \mathbf{y})$, i.e.,*

$$\mathcal{M}_{t-1}(\mathbf{a}_{1:t-1}, \omega) | \mathcal{F}_{t-1}(\mathbf{a}_{1:t-1}, \omega; T, \mathbf{y}) \quad \sim \quad \mathcal{I}(\mathbf{y}_{t-1}(\mathbf{a}_{1:t-1}, \omega; \mathbf{y})). \qquad (126)$$

**Remark 5** (Recursive structure of IRS penalties). *Each of penalty functions (8)–(12) has the following form:*

$$z_t(\mathbf{a}_{1:t}, \omega; T, \mathbf{y}) = \varphi^z(\mathcal{M}_{t-1}(\mathbf{a}_{1:t-1}, \omega), T - t + 1, \mathbf{y}_{t-1}(\mathbf{a}_{1:t-1}, \omega; \mathbf{y})), \qquad (127)$$

*for some function $\varphi^z : \Omega \times \mathbb{N} \times \mathcal{Y} \mapsto \mathbb{R}$, i.e., the penalty at each time is completely determined by the remaining rewards $\mathcal{M}_{t-1}(\mathbf{a}_{1:t-1}, \omega)$, the remaining time horizon $T - t + 1$, and the prior belief $\mathbf{y}_{t-1}(\mathbf{a}_{1:t-1}, \omega)$ at that moment.*

Remark 4 immediately follows from Bayes' rule, and Remark 5 can be easily verified. We observe the recursive structure of the sequential inner problems that the DM solves throughout the decision-making process, which can be characterized by the following property.

**Proposition 6** (Generalized posterior sampling). *For each of penalty functions (8)–(12), the IRS policy $\pi^z$ is randomized in such a way that it takes an action $a$ with the probability that the action $a$ is indeed the best action $a_t^{z,*}$ at that moment, i.e.,*

$$\mathbb{P}\left[ a_t^{\pi^z} = a \,\middle|\, \mathcal{F}_{t-1} \right] = \mathbb{P}\left[ a_t^{z,*} = a \,\middle|\, \mathcal{F}_{t-1} \right], \quad \forall a, \quad \forall t. \qquad (128)$$

*The source of uncertainty in the LHS is the randomness of the policy (embedded in $\tilde{\omega}$) and that in the RHS is the randomness of nature (embedded in $\omega$). We let $a_t^{z,*}$ abbreviate $a_t^{z,*}(\mathbf{a}_{1:t-1}^{\pi^z}, \omega; T, \mathbf{y})$ as defined in (121) and $\mathcal{F}_{t-1}$ abbreviate $\mathcal{F}_{t-1}(\mathbf{a}_{1:t-1}^{\pi^z}, \omega; T, \mathbf{y})$. Here we assume that the tie-breaking rule in $\arg\max$ of (121) is identical to the one used when $\pi^z$ solves the inner problem.*

**Proof.** Fix $t$, $\mathbf{a}_{1:t-1}$ and $\omega$. First, $a_t^{z,*}$ is the best action that maximizes the payoff in the remaining periods:

$$a_t^{z,*}(\mathbf{a}_{1:t-1}, \omega; T, \mathbf{y}) = \underset{a_t'}{\arg\max} \left\{ \max_{\mathbf{a}_{t+1:T}'} \sum_{s=t}^{T} r_s(\mathbf{a}_{1:t-1} \oplus \mathbf{a}_{t:s}', \omega) - z_s(\mathbf{a}_{1:t-1} \oplus \mathbf{a}_{t:s}', \omega; T, \mathbf{y}) \right\}. \qquad (129)$$

By Remark 5, for any $s \in [t, T]$,

$$z_s(\mathbf{a}_{1:t-1} \oplus \mathbf{a}_{t:s}', \omega; T, \mathbf{y}) \qquad (130)$$

$$= \varphi^z(\mathcal{M}_{s-1}(\mathbf{a}_{1:t-1} \oplus \mathbf{a}_{t:s-1}', \omega), T - s + 1, \mathbf{y}_{s-1}(\mathbf{a}_{1:t-1} \oplus \mathbf{a}_{t:s-1}', \omega; \mathbf{y})) \qquad (131)$$

$$= \varphi^z(\mathcal{M}_{s-t}(\mathbf{a}_{t:s-1}', \mathcal{M}_{t-1}(\mathbf{a}_{1:t-1}, \omega)), (T - t + 1) - (s - t), \mathbf{y}_{s-t}(\mathbf{a}_{t:s-1}', \mathcal{M}_{t-1}(\mathbf{a}_{1:t-1}, \omega); \mathbf{y}_{t-1}(\mathbf{a}_{1:t-1}, \omega; \mathbf{y}))) \qquad (132)$$

$$= z_{s-t+1}(\mathbf{a}_{t:s}'; \mathcal{M}_{t-1}(\mathbf{a}_{1:t-1}, \omega), T - t + 1, \mathbf{y}_{t-1}(\mathbf{a}_{1:t-1}, \omega; \mathbf{y})). \qquad (133)$$

For rewards, similarly, we have $r_s(\mathbf{a}_{1:t-1} \oplus \mathbf{a}_{t:s}', \omega) = r_{s-t+1}(\mathbf{a}_{t:s}', \mathcal{M}_{t-1}(\mathbf{a}_{1:t-1}, \omega))$. Therefore, (130) is reformulated as

$$a_t^{z,*} = \underset{a_t'}{\arg\max} \left\{ \max_{\mathbf{a}_{t+1:T}'} \sum_{s=t}^{T} r_{s-t+1}(\mathbf{a}_{t:s}', \mathcal{M}_{t-1}(\mathbf{a}_{1:t-1}, \omega)) - z_{s-t+1}(\mathbf{a}_{t:s}', \mathcal{M}_{t-1}(\mathbf{a}_{1:t-1}, \omega), T - t + 1, \mathbf{y}_{t-1}(\mathbf{a}_{1:t-1}, \omega; \mathbf{y})) \right\}. \qquad (134)$$

Next, consider the IRS policy's action $a_t^{\pi^z}$. It internally solves an instance of the inner problem with the sampled outcome $\tilde{\omega} \sim \mathcal{I}(\mathbf{y}_{t-1}(\mathbf{a}_{1:t-1}, \omega; \mathbf{y}))$, the remaining horizon $T - t + 1$, and the prior belief $\mathbf{y}_{t-1}(\mathbf{a}_{1:t-1}, \omega; \mathbf{y})$:

$$a_t^{\pi^z} = \operatorname*{argmax}_{a_1'} \left\{ \max_{\mathbf{a}_{2:T-t+1}'} \sum_{s=1}^{T-t+1} r_s(\mathbf{a}_{1:s}', \tilde{\omega}) - z_s(\mathbf{a}_{1:s}', \tilde{\omega}, T - t + 1, \mathbf{y}_{t-1}(\mathbf{a}_{1:t-1}, \omega; \mathbf{y})) \right\}. \quad (135)$$

Comparing (134) and (135), we observe that they have the identical functional forms, except that $\mathcal{M}_{t-1}(\mathbf{a}_{1:t-1}, \omega)$ is replaced with $\tilde{\omega}$. Since $\mathcal{M}_{t-1}(\mathbf{a}_{1:t-1}, \omega) | \mathcal{F}_{t-1} \sim \mathcal{I}(\mathbf{y}_{t-1}(\mathbf{a}_{1:t-1}, \omega; \mathbf{y}))$ (Remark 4) and $\tilde{\omega} \sim \mathcal{I}(\mathbf{y}_{t-1}(\mathbf{a}_{1:t-1}, \omega; \mathbf{y}))$, it follows that

$$\mathbb{P}\left[a_t^{z,*}(\mathcal{M}_{t-1}(\mathbf{a}_{1:t-1}, \omega)) = a \,\middle|\, \mathcal{F}_{t-1}\right] = \mathbb{P}\left[a_t^{\pi^z}(\tilde{\omega}) = a \,\middle|\, \mathcal{F}_{t-1}\right]. \quad (136)$$

∎

**Proof sketch of Theorem 3.** For each of penalty functions $z_t^{\text{TS}}$, $z_t^{\text{IRS.FH}}$, and $z_t^{\text{IRS.V-ZERO}}$, we construct a confidence interval process $\{(L_{a,t}, U_{a,t})\}_{a \in \mathcal{A}, t \in [T]}$ such that each of the $(L_{a,t}, U_{a,t})$'s satisfies the following conditions: (i) it is $\mathcal{F}_{t-1}$-measurable and (ii) it regulates the suboptimality of action $a$ at time $t$; more specifically, (ii) means that the following holds with a high probability $1 - \delta$:

$$Q_t^{z,\text{in}}(\mathbf{a}_{1:t-1}, a_t^{z,*}) - Q_t^{z,\text{in}}(\mathbf{a}_{1:t-1}, a_t) \le U_{a_t^{z,*},t} - L_{a_t,t}, \quad \forall a_t \in \mathcal{A}. \quad (**)$$

By Proposition 5,

$$W^z(T, \mathbf{y}) - V(\pi^z, T, \mathbf{y}) \quad (137)$$

$$= \mathbb{E}\left[\sum_{t=1}^{T} Q_t^{z,\text{in}}(\mathbf{a}_{1:t-1}^\pi, a_t^{z,*}) - Q_t^{z,\text{in}}(\mathbf{a}_{1:t-1}^\pi, a_t^\pi)\right] \quad (138)$$

$$\le \mathbb{E}\left[\sum_{t=1}^{T} C \cdot \mathbb{P}_{t-1}[(**) \text{ fails}] + \mathbb{E}_{t-1}\left[Q_t^{z,\text{in}}(\mathbf{a}_{1:t-1}^\pi, a_t^{z,*}) - Q_t^{z,\text{in}}(\mathbf{a}_{1:t-1}^\pi, a_t^\pi)\,\middle|\,(**) \text{ holds}\right]\right] \quad (139)$$

$$\le \mathbb{E}\left[\sum_{t=1}^{T} C\delta + \mathbb{E}_{t-1}\left[U_{a_t^{z,*},t} - L_{a_t^\pi,t}\right]\right] \quad (140)$$

$$= TC\delta + \mathbb{E}\left[\sum_{t=1}^{T} U_{a_t^\pi,t} - L_{a_t^\pi,t}\right], \quad (141)$$

where $C$ is an almost-sure upper bound on instantaneous suboptimality, $\mathbb{P}_{t-1}[\cdot] \triangleq \mathbb{P}[\cdot | \mathcal{F}_{t-1}]$, and $\mathbb{E}_{t-1}[\cdot] \triangleq \mathbb{E}[\cdot | \mathcal{F}_{t-1}]$. The last equality follows from

$$\mathbb{E}_{t-1}\left[U_{a_t^{z,*},t}\right] = \sum_{a=1}^{K} U_{a,t} \times \mathbb{P}_{t-1}\left[a_t^{z,*} = a\right] = \sum_{a=1}^{K} U_{a,t} \times \mathbb{P}_{t-1}\left[a_t^\pi = a\right] = \mathbb{E}_{t-1}\left[U_{a_t^\pi,t}\right], \quad (142)$$

by the predictability of $U_{a,t}$ with respect to $\mathbb{F}$ and Proposition 6. Note that (141) accumulates $U_{a_t^\pi,t} - L_{a_t^\pi,t}$ over $t = 1, \dots, T$, each of which is the length of the confidence interval of the action $a_t^\pi$ taken by the policy at each time. We will show that, whenever the policy plays an arm $a$, the confidence interval of that arm shrinks, and therefore the cumulative suboptimality cannot grow too fast.

**Some facts about the Beta-Bernoulli MAB.** From now on, we restrict our attention to a Beta-Bernoulli MAB in which $\theta_a \sim \text{Beta}(\alpha_a, \beta_a)$ and $R_{a,n} \sim \text{Bernoulli}(\theta_a)$. Recall that after the DM observes the first $n$ reward realizations, the Bayesian updating yields

$$\theta_a | (R_{a,1}, \dots, R_{a,n}) \sim \text{Beta}\left(\alpha_a + \sum_{s=1}^{n} R_{a,n}, \beta_a + n - \sum_{s=1}^{n} R_{a,n}\right), \quad \hat{\mu}_{a,n} = \frac{\alpha_a + \sum_{s=1}^{n} R_{a,s}}{\alpha_a + \beta_a + n}. \quad (143)$$

Note that $\{\hat{\mu}_{a,n}\}_{n \ge 0}$ is a martingale that starts from $\hat{\mu}_{a,0} = \frac{\alpha_a}{\alpha_a + \beta_a}$ and converges to $\lim_{n \to \infty} \hat{\mu}_{a,n} = \theta_a$. Roughly speaking, the (unconditional) distribution of $\hat{\mu}_{a,n}$, starting from a point mass $\frac{\alpha_a}{\alpha_a + \beta_a}$,

diffuses toward Beta$(\alpha_a, \beta_a)$, which is the prior distribution[5] of $\theta_a$. In the following lemma, we characterize the distribution of $\hat{\mu}_{a,n}$ more formally.

**Lemma 2.** *The future Bayesian estimate $\hat{\mu}_{a,n}$ is $\frac{n}{4(\alpha_a+\beta_a)(\alpha_a+\beta_a+n)}$-sub-Gaussian, i.e.,*

$$\mathbb{E}\left[\exp\left(\lambda(\hat{\mu}_{a,n} - \mathbb{E}[\hat{\mu}_{a,n}])\right)\right] \leq \exp\left(\frac{\lambda^2}{2} \times \frac{n}{4(\alpha_a + \beta_a)(\alpha_a + \beta_a + n)}\right), \quad \forall \lambda \in \mathbb{R}. \quad (144)$$

**Proof.** Since (i) $\mathbb{E}[\hat{\mu}_{a,n}] = \hat{\mu}_{a,0} = \frac{\alpha_a}{\alpha_a+\beta_a}$, (ii) $R_{a,n}$'s are i.i.d. conditioned on $\theta_a$, (iii) Bernoulli$(\theta_a)$ is $\frac{1}{4}$-sub-Gaussian (for any $\theta_a$), and (iv) Beta$(\alpha, \beta)$ is $\frac{1}{4(\alpha+\beta+1)}$-sub-Gaussian [16], it follows that, for any $\lambda \in \mathbb{R}$,

$$\mathbb{E}\left[\exp\left(\lambda(\hat{\mu}_{a,n} - \hat{\mu}_{a,0})\right)\right] \quad (145)$$

$$= \mathbb{E}\left[\exp\left(\frac{\lambda}{\alpha_a + \beta_a + n} \times \left((\alpha_a + \sum_{s=1}^{n} R_{a,s}) - (\alpha_a + \beta_a + n)\hat{\mu}_{a,0}\right)\right)\right] \quad (146)$$

$$\overset{(i)}{=} \mathbb{E}\left[\exp\left(\frac{\lambda}{\alpha_a + \beta_a + n} \times \left(\sum_{s=1}^{n}(R_{a,s} - \theta_a) + n \cdot (\theta_a - \hat{\mu}_{a,0})\right)\right)\right] \quad (147)$$

$$= \mathbb{E}\left[\mathbb{E}\left\{\exp\left(\frac{\lambda}{\alpha_a + \beta_a + n} \times \sum_{s=1}^{n}(R_{a,s} - \theta_a)\right)\middle| \theta_a\right\} \times \exp\left(\frac{\lambda}{\alpha_a + \beta_a + n} \times n \cdot (\theta_a - \hat{\mu}_{a,0})\right)\right] \quad (148)$$

$$\overset{(ii)}{=} \mathbb{E}\left[\mathbb{E}\left\{\exp\left(\frac{\lambda}{\alpha_a + \beta_a + n} \times (R_{a,1} - \theta_a)\right)\middle| \theta_a\right\}^n \times \exp\left(\frac{\lambda}{\alpha_a + \beta_a + n} \times n \cdot (\theta_a - \hat{\mu}_{a,0})\right)\right] \quad (149)$$

$$\overset{(iii)}{\leq} \mathbb{E}\left[\left\{\exp\left(\frac{\lambda^2}{2(\alpha_a + \beta_a + n)^2} \times \frac{1}{4}\right)\right\}^n \times \exp\left(\frac{\lambda}{\alpha_a + \beta_a + n} \times n \cdot (\theta_a - \hat{\mu}_{a,0})\right)\right] \quad (150)$$

$$= \exp\left(\frac{\lambda^2}{2} \times \frac{n}{4(\alpha_a + \beta_a + n)^2}\right) \times \mathbb{E}\left[\exp\left(\frac{\lambda n}{\alpha_a + \beta_a + n} \times (\theta_a - \hat{\mu}_{a,0})\right)\right] \quad (151)$$

$$\overset{(iv)}{\leq} \exp\left(\frac{\lambda^2}{2} \times \frac{n}{4(\alpha_a + \beta_a + n)^2}\right) \times \exp\left(\frac{\lambda^2 n^2}{2(\alpha_a + \beta_a + n)^2} \times \frac{1}{4(\alpha_a + \beta_a + 1)}\right) \quad (152)$$

$$\leq \exp\left(\frac{\lambda^2}{2} \times \frac{n}{4(\alpha_a + \beta_a + n)^2}\right) \times \exp\left(\frac{\lambda^2 n^2}{2(\alpha_a + \beta_a + n)^2} \times \frac{1}{4(\alpha_a + \beta_a)}\right) \quad (153)$$

$$= \exp\left(\frac{\lambda^2}{2} \times \frac{n(\alpha_a + \beta_a) + n^2}{4(\alpha_a + \beta_a + n)^2(\alpha_a + \beta_a)}\right) = \exp\left(\frac{\lambda^2}{2} \times \frac{n}{4(\alpha_a + \beta_a + n)(\alpha_a + \beta_a)}\right). \quad (154)$$

∎

**(1) Suboptimality analysis of TS** (20). Define

$$\Delta_{a,t} \triangleq \sqrt{\frac{\log T}{n_{t-1}^{\pi}(a)}}, \quad U_{a,t} \triangleq \min\left\{\hat{\mu}_{a,n_{t-1}^{\pi}(a)} + \Delta_{a,t}, 1\right\}, \quad L_{a,t} \triangleq \max\left\{\hat{\mu}_{a,n_{t-1}^{\pi}(a)} - \Delta_{a,t}, 0\right\}, \quad (155)$$

where $n_{t-1}^{\pi}(a) \triangleq n_{t-1}(\mathbf{a}_{1:t-1}^{\pi}, a)$ represents how many times the policy $\pi$ has pulled an arm $a$ before time $t$. The confidence interval $(L_{a,t}, U_{a,t})$ constructs the high probability lower/upper bounds on $\mu_a(\theta_a) \,(= \theta_a)$ at time $t$ and it is $\mathcal{F}_{t-1}$-measurable. Conditioned on $\mathcal{F}_{t-1}$, $\mu_a(\theta_a)$ is distributed with Beta$(\alpha_a + \sum_{s=1}^{n_{t-1}^{\pi}(a)} R_{a,s}, \beta_a + n_{t-1}^{\pi}(a) - \sum_{s=1}^{n_{t-1}^{\pi}(a)} R_{a,s})$, which is $\frac{1}{4(\alpha_a + \beta_a + n_{t-1}^{\pi}(a) + 1)}$-sub-Gaussian. By Chernoff's inequality,

$$\mathbb{P}_{t-1}\left[\mu_a(\theta_a) \geq U_{a,t}\right] = \mathbb{P}_{t-1}\left[\mu_a(\theta_a) - \hat{\mu}_{a,n_{t-1}^{\pi}(a)} \geq \Delta_{a,t}\right] \quad (156)$$

$$\leq \exp\left(-\frac{\Delta_{a,t}^2}{2 \times \left(4(\alpha_a + \beta_a + n_{t-1}^{\pi}(a) + 1)\right)^{-1}}\right) \quad (157)$$

$$\leq \exp\left(-2n_{t-1}^{\pi}(a) \times \frac{\log T}{n_{t-1}^{\pi}(a)}\right) = \frac{1}{T^2}. \quad (158)$$

Similarly, we have $\mathbb{P}_{t-1}\left[\mu_a(\theta_a) \leq L_{a,t}\right] \leq \frac{1}{T^2}$. We define an event $\mathcal{E}$ in which $(L_{a,t}, U_{a,t})$ is indeed a valid confidence interval for every arm $a$ at every time $t$:

$$\mathcal{E} \triangleq \{\mu_a(\theta_a) \in (L_{a,t}, U_{a,t}), \quad \forall a, \quad \forall t\}. \tag{159}$$

By the above concentration inequalities, the sequence of confidence intervals contains the true mean $\mu_a(\theta_a)$ with a very high probability:

$$\mathbb{P}\left[\mathcal{E}^c\right] \leq \mathbb{E}\left[\sum_{a=1}^{K}\sum_{t=1}^{T} \mathbb{P}_{t-1}\left[\mu_a(\theta_a) \geq U_{a,t}\right] + \mathbb{P}_{t-1}\left[\mu_a(\theta_a) \leq L_{a,t}\right]\right] \leq \frac{2K}{T}. \tag{160}$$

With $z_t^{\text{TS}}$, the Q-value of the inner problem is

$$Q_t^{z,\text{in}}(\mathbf{a}_{1:t-1}, a_t) = \mu_{a_t}(\theta_{a_t}) + (T-t) \times \mu_{a_t^*}(\theta_{a_t^*}). \tag{161}$$

Given the event $\mathcal{E}$, in which $\mu_a(\theta_a) \in (L_{a,t}, U_{a,t})$ for all $a$, we have

$$Q_t^{z,\text{in}}(\mathbf{a}_{1:t-1}, a_t^*) - Q_t^{z,\text{in}}(\mathbf{a}_{1:t-1}, a_t) = \mu_{a_t^*}(\theta_{a_t^*}) - \mu_{a_t}(\theta_{a_t}) \leq U_{a_t^*,t} - L_{a_t,t}. \tag{162}$$

As outlined earlier, the total suboptimality of $\pi^{\text{TS}}$ is limited by

$$W^{\text{TS}}(T, \mathbf{y}) - V(\pi^{\text{TS}}, T, \mathbf{y}) \leq T \times \mathbb{P}\left[\mathcal{E}^c\right] + \mathbb{E}\left[\sum_{t=1}^{T} U_{a_t^\pi, t} - L_{a_t^\pi, t}\right] \leq 2K + \mathbb{E}\left[\sum_{a=1}^{K}\sum_{t=1}^{T} \min(1, 2\Delta_{a,t}) \cdot \mathbf{1}\{a_t^\pi = a\}\right]. \tag{163}$$

For each arm $a = 1, \ldots, K$,

$$\sum_{t=1}^{T} \min(1, 2\Delta_{a,t}) \cdot \mathbf{1}\{a_t^\pi = a\} \leq 1 + \sum_{n=2}^{n_T^\pi(a)} 2\sqrt{\frac{\log T}{n-1}} \leq 1 + 2\sqrt{\log T} \times \int_{x=0}^{n_T^\pi(a)} \frac{dx}{\sqrt{x}} \leq 1 + 4\sqrt{\log T} \times \sqrt{n_T^\pi(a)}. \tag{164}$$

By the Cauchy–Schwartz inequality and since $\sum_{a=1}^{K} n_T^\pi(a) = T$,

$$\sum_{a=1}^{K}\left(1 + 4\sqrt{\log T} \times \sqrt{n_T^\pi(a)}\right) \leq K + 4\sqrt{\log T} \times \sqrt{K \sum_{a=1}^{K} n_T^\pi(a)} = K + 4\sqrt{\log T} \times \sqrt{KT}. \tag{165}$$

Combining all the results, we obtain

$$W^{\text{TS}}(T, \mathbf{y}) - V(\pi^{\text{TS}}, T, \mathbf{y}) \leq 3K + 4\sqrt{\log T} \times \sqrt{KT}. \tag{166}$$

∎

## (2) Suboptimality analysis of IRS.FH (21). Note that $z_t^{\text{IRS.FH}}$ yields

$$Q_t^{z,\text{in}}(\mathbf{a}_{1:t-1}, a_t^*) - Q_t^{z,\text{in}}(\mathbf{a}_{1:t-1}, a_t) = \hat{\mu}_{a_t^*, n_{t-1}^\pi(a_t^*)+T-t} - \hat{\mu}_{a_t, n_{t-1}^\pi(a_t)+T-t}. \tag{167}$$

When $t = 1$, $\hat{\mu}_{a, n_{t-1}^\pi(a)+T-t}$ coincides with $\hat{\mu}_{a, T-1}$. We need to bound $\hat{\mu}_{a_t^*, n_{t-1}^\pi(a_t^*)+T-t}$ instead of $\mu_a(\theta_a)$. Note that, conditioned on $\mathcal{F}_{t-1}$, $\{\hat{\mu}_{a, n_{t-1}^\pi(a)+n}\}_{n \geq 0}$ is a martingale whose distribution starts from a point mass $\hat{\mu}_{a, n_{t-1}^\pi(a)}$ and diffuses toward the prior distribution $\text{Beta}(\alpha_a + \sum_{s=1}^{n_{t-1}^\pi(a)} R_{a,s}, \beta_a + n_{t-1}^\pi(a) - \sum_{s=1}^{n_{t-1}^\pi(a)} R_{a,s})$. For any $a$ and $n \geq 0$, by Lemma 2, we have

$$\mathbb{E}_{t-1}\left[\exp(\lambda(\hat{\mu}_{a, n_{t-1}^\pi(a)+n} - \hat{\mu}_{a, n_{t-1}^\pi(a)}))\right] \leq \exp\left(\frac{\lambda^2}{2} \times \frac{n}{4(\alpha_a + \beta_a + n_{t-1}^\pi(a))(\alpha_a + \beta_a + n_{t-1}^\pi(a) + n)}\right) \tag{168}$$

$$\leq \exp\left(\frac{\lambda^2}{2} \times \frac{n}{4n_{t-1}^\pi(a)(n_{t-1}^\pi(a) + n)}\right). \tag{169}$$

With $n = T - t$, we can conclude that $\hat{\mu}_{a_t^*, n_{t-1}^\pi(a_t^*)+T-t}$ is $\frac{T-t}{4n_{t-1}^\pi(a)(T-t+n_{t-1}^\pi(a))}$-sub-Gaussian.

Define

$$\Delta_{a,t} \triangleq \sqrt{\frac{T-t}{n_{t-1}^\pi(a) + T - t} \times \frac{\log T}{n_{t-1}^\pi(a)}}, \quad U_{a,t} \triangleq \min\left\{\hat{\mu}_{a, n_{t-1}^\pi(a)} + \Delta_{a,t}, 1\right\}, \quad L_{a,t} \triangleq \max\left\{\hat{\mu}_{a, n_{t-1}^\pi(a)} - \Delta_{a,t}, 0\right\}. \tag{170}$$

By Chernoff's inequality,

$$\mathbb{P}_{t-1}\left[\hat{\mu}_{a_t^*, n_{t-1}^\pi(a_t^*)+T-t} \geq U_{a,t}\right] = \mathbb{P}_{t-1}\left[\hat{\mu}_{a, T-1} - \hat{\mu}_{a, n_{t-1}^\pi(a)} \geq \Delta_{a,t}\right] \tag{171}$$

$$\leq \exp\left(-\frac{\Delta_{a,t}^2}{2 \times \frac{T-t}{4n_{t-1}^\pi(a)(T-t+n_{t-1}^\pi(a))}}\right) = \exp\left(-2\log T\right) = \frac{1}{T^2}. \tag{172}$$

Similarly, we can show that $\mathbb{P}_{t-1}\left[\hat{\mu}_{a_t^*, n_{t-1}^\pi(a_t^*)+T-t} \le L_{a,t}\right] \le \frac{1}{T^2}$.

Analogously to the proof of TS, we can show that

$$W^{\mathrm{TS}}(T, \mathbf{y}) - V(\pi^{\mathrm{TS}}, T, \mathbf{y}) \le 2K + \mathbb{E}\left[\sum_{a=1}^{K}\sum_{t=1}^{T} \min(1, 2\Delta_{a,t}) \cdot \mathbf{1}\{a_t^\pi = a\}\right]. \tag{173}$$

Since $n_{t-1}^\pi(a) \le t$, we have $\frac{T-t}{n_{t-1}^\pi(a)+T-t} = \left(1 + \frac{n_{t-1}^\pi(a)}{T-t}\right)^{-1} \le \left(1 + \frac{n_{t-1}^\pi(a)}{T-n_{t-1}^\pi(a)}\right)^{-1} = 1 - \frac{n_{t-1}^\pi(a)}{T}$ and

$$\Delta_{a,t} \le \sqrt{\left(1 - \frac{n_{t-1}^\pi(a)}{T}\right) \times \frac{\log T}{n_{t-1}^\pi(a)}} = \sqrt{\log T} \times \sqrt{\frac{1}{n_{t-1}^\pi(a)} - \frac{1}{T}} \le \sqrt{\log T} \times \left(\frac{1}{\sqrt{n_{t-1}^\pi(a)}} - \frac{\sqrt{n_{t-1}^\pi(a)}}{2T}\right). \tag{174}$$

Consequently, for each $a$,

$$\sum_{t=1}^{T} \min(1, 2\Delta_{a,t}) \cdot \mathbf{1}\{a_t^\pi = a\} \le 1 + 2\sqrt{\log T} \times \sum_{n=2}^{n_T^\pi(a)} \left(\frac{1}{\sqrt{n-1}} - \frac{\sqrt{n-1}}{2T}\right) \tag{175}$$

$$\le 1 + 2\sqrt{\log T} \times \int_{x=0}^{n_T^\pi(a)} \left(\frac{1}{\sqrt{x}} - \frac{\sqrt{x}}{2T}\right) \tag{176}$$

$$= 1 + 2\sqrt{\log T} \times \left(2\sqrt{n_T^\pi(a)} - \frac{(n_T^\pi(a))^{3/2}}{3T}\right). \tag{177}$$

Note that, since $x \mapsto x^{3/2}$ is a convex function and $\sum_{a=1}^{K} n_T^\pi(a) = T$,

$$\sum_{a=1}^{K} (n_T^\pi(a))^{3/2} \ge \sum_{a=1}^{K} \left(\frac{T}{K}\right)^{3/2} = \sqrt{T^3/K}. \tag{178}$$

By the Cauchy–Schwarz inequality, as in TS, we have $\sum_{a=1}^{K} \sqrt{n_T^\pi(a)} \le \sqrt{KT}$. As a result,

$$W^{\mathrm{TS}}(T, \mathbf{y}) - V(\pi^{\mathrm{TS}}, T, \mathbf{y}) \le 2K + \mathbb{E}\left[\sum_{a=1}^{K} 1 + 2\sqrt{\log T} \times \left(2\sqrt{n_T^\pi(a)} - \frac{(n_T^\pi(a))^{3/2}}{3T}\right)\right] \tag{179}$$

$$\le 3K + 2\sqrt{\log T} \times \left(2\sqrt{KT} - \frac{1}{3}\sqrt{T/K}\right). \tag{180}$$

$\blacksquare$

**(3) Suboptimality analysis of IRS.V-ZERO (22).** Consider an optimal allocation $\mathbf{n}^*$ of the inner problem of IRS.V-ZERO when the remaining time is $T$. For an arm $a$ on which the optimal solution allocates at least one pull, i.e., $n^*(a) > 0$, a policy does not incur suboptimality by pulling the arm $a$ (the arms that $n^*(a) > 0$ are all optimal and their Q-values tie). A policy incurs suboptimality only when pulling an arm $a$ such that $n^*(a) = 0$, in which case we lose $\min_{a':n^*(a')>0}\{\hat{\mu}_{a', n^*(a')-1}\} - \hat{\mu}_{a,0}$ (we lose the last pull of one of the optimal arms) where the term $\min_{a':n^*(a')>0}\{\hat{\mu}_{a', n^*(a')-1}\}$ is limited by $\max_{0 \le n \le T-1} \hat{\mu}_{a^*, n}$ for some $a^*$ such that $n^*(a^*) > 0$. Extending this argument, at a certain time $t$, when the remaining time is $T - t + 1$, we have

$$Q_t^{z,\mathrm{in}}(\mathbf{a}_{1:t-1}, a_t^*) - Q_t^{z,\mathrm{in}}(\mathbf{a}_{1:t-1}, a_t) \le \max_{0 \le n \le T-t}\left\{\hat{\mu}_{a_t^*, n_{t-1}^\pi(a_t^*)+n}\right\} - \hat{\mu}_{a_t, n_{t-1}^\pi(a_t)}. \tag{181}$$

We need to regulate $\max_{0 \le n \le T-t}\left\{\hat{\mu}_{a_t^*, n_{t-1}^\pi(a_t^*)+n}\right\}$. As before, we define

$$\Delta_{a,t} \triangleq \sqrt{\frac{T-t}{n_{t-1}^\pi(a)+T-t} \times \frac{\log T}{n_{t-1}^\pi(a)}}, \quad U_{a,t} \triangleq \min\left\{\hat{\mu}_{a,n_{t-1}^\pi(a)} + \Delta_{a,t}, 1\right\}, \quad L_{a,t} \triangleq \hat{\mu}_{a,n_{t-1}^\pi(a)}. \tag{182}$$

Note that we take $L_{a,t}$ that are different from those in the previous case, but still $\mathcal{F}_{t-1}$-measurable. Given that $\{\hat{\mu}_{a,n_{t-1}^\pi(a)+n} - \hat{\mu}_{a,n_{t-1}^\pi(a)}\}_{n \ge 0}$ is a martingale, $\left\{\exp\left(\lambda(\hat{\mu}_{a,n_{t-1}^\pi(a)+n} - \hat{\mu}_{a,n_{t-1}^\pi(a)})\right)\right\}_{n \ge 0}$ is a non-negative supermartingale due to the convexity of $\exp(\cdot)$. By Doob's maximal inequality and Lemma 2, for any

$\lambda \geq 0$,

$$\mathbb{P}_{t-1}\left[\max_{0 \leq n \leq T-t}\left\{\hat{\mu}_{a,n_{t-1}^{\pi}(a)+n}\right\} \geq U_{a,t}\right] = \mathbb{P}_{t-1}\left[\max_{0 \leq n \leq T-t}\left\{\hat{\mu}_{a,n_{t-1}^{\pi}(a)+n} - \hat{\mu}_{a,n_{t-1}^{\pi}(a)}\right\} \geq \Delta_{a,t}\right]$$
(183)

$$\leq \mathbb{P}_{t-1}\left[\max_{0 \leq n \leq T-t}\left\{\exp\left(\lambda(\hat{\mu}_{a,n_{t-1}^{\pi}(a)+n} - \hat{\mu}_{a,n_{t-1}^{\pi}(a)})\right)\right\} \geq \exp\left(\lambda\Delta_{a,t}\right)\right]$$
(184)

$$\leq \frac{\mathbb{E}_{t-1}\left[\exp\left(\lambda(\hat{\mu}_{a,n_{t-1}^{\pi}(a)+T-t} - \hat{\mu}_{a,n_{t-1}^{\pi}(a)})\right)\right]}{\exp(\lambda\Delta_{a,t})}$$
(185)

$$\leq \exp\left(\frac{\lambda^2}{2} \times \frac{T-t}{4n_{t-1}^{\pi}(a)(n_{t-1}^{\pi}(a)+T-t)} - \lambda\Delta_{a,t}\right).$$
(186)

For $\lambda$ that minimizes the RHS and $\Delta_{a,t}$ that is defined above, we have

$$\mathbb{P}_{t-1}\left[\max_{0 \leq n \leq T-t}\left\{\hat{\mu}_{a,n_{t-1}^{\pi}(a)+n}\right\} \geq U_{a,t}\right] \leq \exp\left(-\frac{2n_{t-1}^{\pi}(a)(n_{t-1}^{\pi}(a)+T-t)}{T-t} \times \Delta_{a,t}^2\right) = \frac{1}{T^2}.$$
(187)

Note that $\max_{0 \leq n \leq T-t}\left\{\hat{\mu}_{a,n_{t-1}^{\pi}(a)+n}\right\} \geq L_{a,t} \equiv \hat{\mu}_{a,n_{t-1}^{\pi}(a)}$ by defintion. We have shown that

$$\mathbb{P}\left[\mathcal{E} \triangleq \left\{\max_{0 \leq n \leq T-t}\left\{\hat{\mu}_{a,n_{t-1}^{\pi}(a)+n}\right\} \in [L_{a,t}, U_{a,t}), \quad \forall a, \forall t\right\}\right] \geq 1 - \frac{K}{T}.$$
(188)

Therefore, using the facts derived for TS and IRS.FH, we obtain

$$W^{\mathrm{TS}}(T,\mathbf{y}) - V(\pi^{\mathrm{TS}}, T, \mathbf{y}) \leq T\mathbb{P}[\mathcal{E}^c] + \mathbb{E}\left[\sum_{t=1}^{T} U_{a_t^{\pi},t} - L_{a_t^{\pi},t}\right]$$
(189)

$$\leq K + \mathbb{E}\left[\sum_{a=1}^{K}\sum_{t=1}^{T} \min(1, \Delta_{a,t})\mathbf{1}\{a_t^{\pi} = a\}\right]$$
(190)

$$\leq 2K + \sqrt{\log T} \times \left(2\sqrt{KT} - \frac{1}{3}\sqrt{T/K}\right).$$
(191)

∎

## Footnotes

[3] For details, see the proof of the strong duality theorem in §C.1. While the maximal value does not depend on $\omega$, the optimal action sequence still depends on $\omega$. More specifically, it is the sequence of actions that the (non-anticipating) Bayesian optimal policy will take when $\omega$ is sequentially revealed.

[4]In case of IRS.V-ZERO, we select the arm with the largest pull allocation as a first action.

[5] Conditioned on $\theta_a$, $\{\hat{\mu}_{a,n}\}_{n\geq 0}$ is no longer a martingale and the distribution of $\hat{\mu}_{a,n}$ starts from a point mass $\frac{\alpha_a}{\alpha_a+\beta_a}$, diffuses for a while, and ends up at a point mass $\theta_a$. With the randomness of $\theta_a$, $\{\hat{\mu}_{a,n}\}_{n\geq 0}$ is a martingale and the distribution of $\hat{\mu}_{a,n}$ gets wider as $n$ increases.