[Reviews · NeurIPS 2019]

Reviewer 1



This work studies a Bayesian multi-armed bandit with a known and finite time horizon. The proposed framework introduces a technique of relaxing the non-anticipativity constraint in operations research to bandit problems and generates several interesting algorithms. These algorithms can exploit the knowledge of time horizon in a systematic way and achieves better performance than thompson sampling. The paper is well-written and organized. And I believe the results are very interesting to the bandit community and even larger field of reinforcement learning. I have some comments: 1) Can this framework get rid of the knowledge of T? In the frequentist setting, one can use a doubling trick to extend the regret bound obtained in knowing T to the setting of not knowing T. So I am wondering this framework can be extended to the setting of not knowing T. That would be interesting in practice. 2) In most cases, IRS has computational complexity that is linear (or even polynomial) in T for each round. This complexity can be a problem in some cases when the time horizon is millions or more, which could be limit to its practical value. 3) Any theoretical results for IRS.V-EMAX in the form of Theorem 2 and 3? a minor one: line 34: focus "on" the Bayesian setting ... ==== after response==== Rebuttal read. Keep my score.

Reviewer 2



The paper applies information relaxation techniques to the finite-horizon multi-armed bandit problem. By proposing a series of information relaxation penalty functions, the authors are able to derive a series of upper bounds on the optimal value function that are increasingly tighter at the cost of increasing computational complexity. The authors also propose a spectrum of algorithms with increasing computational complexity using these penalty functions, with one end being Thompson sampling and the other end being the optimal policy. Algorithms derived from this technique take into account the effect of having a finite horizon in a natural way. I think these results are very interesting and the approach shows great promise for dealing with problems with a relatively short time horizon. The simulation results show that the proposed algorithms perform better than vanilla Thompson sampling, with the cost of more compute time. What seems interesting to me is that the performance of information directed sampling looks comparable to IRS algorithms, even though IDS only looks at the current time period at each time, and does not consider the future or having a finite horizon. What is the advantage of IRS over IDS? My guess is that IDS probably does not scale well with number of arms. Could you please confirm? The paper is pretty well-written in general. There are a few minor typos, so please proofread. Also, since IRS.Index achieves the best regret empirically, it might be worthwhile to discuss it in the main text.

Reviewer 3



I think this is a good paper with a clear structure and is nicely written. However, the paper focuses on stochastic bandits with independent arms, with is an extensively studied realm with abundant results. Although the authors present a convincing theoretical analysis, there is no significant improvement over previous bounds (as the authors mention, TS for MAB is already minimax optimal). The generalisation of TS to finite-horizon problems is not novel, and the exploration-exploitation trade-off has also been addressed in previous literature. This could be the start of an interesting line of work, but I do not think the current version is above the acceptation threshold.

[Author Response · NeurIPS 2019]

We appreciate the careful reviews!

**Reviewer 1:** (1) "Can this framework get rid of the knowledge of $T$?" Some information about $T$ is necessary, since
the time horizon is an important ingredient in optimally balancing exploration and exploitation. That said, precise
knowledge of $T$ is not necessary. The framework (penalties, policies, and upper bounds) can naturally incorporate
the unknown $T$ within the *Bayesian setting*: i.e., the horizon $T$ is also a random variable whose prior distribution is
known. As a simple case, if $T$ is independent of the DM's actions, we can reformulate the objective function of the inner
problem as $\sum_{t=1}^{\infty} \gamma_t(r_t(\mathbf{a}_{1:t}; \omega) - z_t(\mathbf{a}_{1:t}; \omega))$ where the discount factor $\gamma_t \triangleq \mathbb{P}[T \geq t]$ is the survivor probability, and
$r_t(\cdot)$ and $z_t(\cdot)$ are the reward and penalty terms used in the paper. Alternatively, we can treat the random variable $T$ like
the random reward realizations – sample $T$ from its prior distribution while a penalty function (additionally) penalizes
for the gain from knowing $T$ (you can imagine that the outcome $\omega$ now includes the realization of $T$). Structural results
such as weak duality and strong duality will continue to hold.

(2) "In most cases, IRS has computational complexity that is linear (or even polynomial) in $T$ for each round." To
be fair, the computational complexity of IRS.FH is independent of $T$ (like TS). Moreover, we can construct a dual
feasible penalty function that mixes IRS.FH and IRS.V-ZERO, which induces an algorithm whose complexity is
$O\left(K \min\{T, T_0\}^2\right)$ for some predefined constant $T_0$ (in the inner problem, IRS.V-ZERO-like penalties are applied
for the initial $\lfloor T_0/K \rfloor$ pulls and then IRS.FH-like penalties are applied for the later pulls). Such a practical variant
has performance that does not scale with $T$ beyond $T_0$. In our experiments, this variant works well even for moderate
values of $T_0$. Other heuristic variations of IRS policies with bounded $T$ dependence are also possible.

(3) "Any theoretical results for IRS.V-EMAX?" We have the additional result that $W^{\text{IRS.V-EMAX}}(T, \mathbf{y}) \leq$
$W^{\text{TS}}(T, \mathbf{y})$, this could be added to Theorem 2. We don't have a suboptimality analysis for IRS.V-EMAX yet.

**Reviewer 2: Comparison with information-directed sampling (IDS).** It is also remarkable to us that IDS performs
well without the knowledge of $T$. The IRS policies outperform when the finite horizon creates a considerable tension in
the exploitation-exploration trade-off. As illustrated in the numerical experiments, this would be the case when the
arms are dissimilar and the time horizon is short relative to the number of arms.

We believe that our algorithms have other advantages over IDS. First, IDS requires a significant amount of work
that is specific to the application; e.g., it requires to compute the expected change in entropy, which is typically
obtained by a numerical integration after doing some distribution-specific reformulation. For example, if some arms
yield normally-distributed rewards and the others yield Bernoulli-distributed rewards, implementing IDS will be very
challenging. Moreover, IDS's computational complexity is $O(K^2)$ per decision, whereas IRS.FH and IRS.V-ZERO are
linear in $K$. Finally, this framework can naturally deal with other constraints apart from the time-horizon one; see the
answer for Reviewer 3 below.

**Reviewer 3: Additional discussion on our contributions.** Respectfully, we believe that our generalization of TS
to finite-horizon problems is novel and has not appeared in the literature. A common heuristic for the finite-horizon
setting would be *posterior reshaping*, mentioned in Chapelle and Li (2011), which reduces the variance of the posterior
distribution with an ad hoc parameter. Another relevant work is Russo, Tse and Van Roy (2017), in which the authors
propose *satisficing Thompson sampling* for the discounted infinite-horizon setting, which also introduces an auxiliary
parameter to control the degree of exploitation explicitly. In contrast to these heuristic proposals, this paper provides a
principled method that does not require any tuning or additional parameters, and suggests a unified framework that
includes TS and the Bayesian optimal policy as special cases. Also note that the decision making procedure of every
IRS policy is recursive like TS: i.e., the decision at a certain moment depends only on the posterior distribution and the
remaining horizon at that moment.

We absolutely agree with the fact that the stochastic MAB with independent arms has already been studied extensively.
That said, to the extent that this problem is practically interesting, we provide methods that are competitive with
commonly employed solution methods such as TS. Moreover, even though this paper focuses on this simplest setting,
our framework applies for more complicated settings. Consider the following examples: (a) Correlated arms in a
finite-horizon setting (e.g., $R_{a,n} \sim \mathcal{N}(\mathbf{x}_a^\top \boldsymbol{\theta}, \sigma^2)$ where $\boldsymbol{\theta}$ is shared across the arms): IRS.V-ZERO can be immediately
implemented by adopting the DP algorithm discussed in §B.2. (b) MAB with the delayed reward realization: IRS.FH
can be immediately implemented by simulating the DM's learning process in the presence of delay. (c) MAB with a
budget constraint (in which each arm consumes a certain amount of budget and the DM wants to maximize the total
reward within a limited budget): all IRS algorithms can be implemented by solving a budget-constrained optimization
problem instead of a horizon-constrained optimization problem. In these extensions, we obtain not only the online
decision making policies but also their performance bounds as in this paper. Generally speaking, our framework
provides a systemic way of improving TS by taking into account the exploitation-exploration trade-off more carefully,
particularly in the presence of some constraint that incurs incomplete learning. We believe that this feature is novel in
the literature and also very crucial in practice.

[Meta-Review · NeurIPS 2019]

The paper makes a conceptual algorithmic contribution to sequential decision making under uncertainty. It proposes what appears to be a new way of looking at sampling based multi armed bandit algorithms like Thompson sampling, and unifies it with a host of other methods going all the way to the (intractable) Bayes-optimal sequential bandit algorithm. All reviewers seemed to agree on the conceptual value that the new viewpoint brings to algorithm design and analysis, although a crisp consensus could not be reached amongst them. Nevertheless, the discussion post author feedback saw a majority of the reviewers championing the paper's contribution, which justifies accepting the paper for presentation.